# Flexible hyperspectral surface plasmon resonance microscopy

Ziwei Liu [1,2], Jingning Wu [1,2], Chen Cai [1,2], Bo Yang [1,2] & Zhi-mei Qi [1,2,3] ✉

Optical techniques for visualization and quantification of chemical and biological analytes are always highly desirable. Here we show a hyperspectral surface plasmon resonance microscopy (HSPRM) system that uses a hyperspectral microscope to analyze the selected area of SPR image produced by a prism-based spectral SPR sensor. The HSPRM system enables monochromatic and polychromatic SPR imaging and single-pixel spectral SPR sensing, as well as two-dimensional quantification of thin films with the measured resonance-wavelength images. We performed pixel-by-pixel calibration of the incident angle to remove pixel-to-pixel differences in SPR sensitivity, and demonstrated the HSPRM's capabilities by using it to quantify monolayer graphene thickness distribution, inhomogeneous protein adsorption and single-cell adhesion. The HSPRM system has a wide spectral range from 400 nm to 1000 nm, an optional field of view from 0.884 mm$^2$ to 0.003 mm$^2$ and a high lateral resolution of 1.2 μm, demonstrating an innovative breakthrough in SPR sensor technology.

Surface plasmon resonance imaging (SPRi) is a spatially resolved evanescent sensing technique that is highly sensitive to changes in refractive index (RI) and/or thickness near the metal/dielectric interface along which the plasmon wave propagates. Because chemical and biological reactions at the surface and interface are always accompanied by changes in RI and/or thickness, SPRi sensors have the excellent capability for label-free visualization and high-throughput detection of chemical and biological measurands. Since the birth of the first SPRi sensor in 1987[1], this type of evanescent sensor has been extensively studied with a focus on improving its performance and expanding its field of application. On the other hand, the performance improvement of SPRi sensors is also related with the rapid development of advanced digital camera technology. For example, the fast data processing speed and high pixel density of current digital cameras enable SPRi sensors to track multiple biochemical reaction processes in parallel with high time resolution and high image resolution. SPRi sensors have been commercialized in many companies, mainly including GWC Technologies (USA), IBIS Technologies (Netherlands), and Horiba (France). These commercial SPRi sensors are now playing an important role in various practical applications as the mainstream label-free biochemical analyzers.

SPRi sensors can be constructed with either a prism coupler or an objective with high numerical aperture (NA). Compared with objective-based SPRi sensors, prism-based SPRi sensors have a larger field of view (FOV) but lower lateral resolution. The existing SPRi sensors typically use a laser as the light source, and they operate at a fixed angle of incidence and are initially in resonance to generate a dark-field background. Loading of the sample on the SPR chip causes the resonance to deviate, resulting in a greyscale image of the sample. The intensity at a single pixel of the image varies within a limited range with the sample RI and/or thickness and can easily reach saturation. This means that the detection dynamic range of the monochromatic SPRi sensors is very narrow, typically 0.02 RIU[2], which makes it impossible for the sensor to simultaneously visualize multiple objects with large differences in RI and/or size. SPR sensors based on either angle interrogation or wavelength interrogation allow quantification of sample RI and/or thickness by fitting the measured resonance angle or measured resonance wavelength (RW) using the Fresnel equations. This capability, however, is lost in the monochromatic SPRi sensors because the intensity measured at each pixel of the SPR image is affected by multiple factors, including the power of the light source, the coupling efficiency, and the detector sensitivity, not only the sample properties.

[1]State Key Laboratory of Transducer Technology, Aerospace Information Research Institute, Chinese Academy of Sciences, Beijing 100190, China. [2]School of Electronic, Electrical, and Communication Engineering, University of Chinese Academy of Sciences, Beijing 100049, China. [3]School of Optoelectronics, University of Chinese Academy of Sciences, Beijing 100049, China. ✉e-mail: zhimei-qi@mail.ie.ac.cn

Spectral SPRi sensors with broadband light sources can provide full-color images where different objects appear in different colors[3]. Compared to monochromatic SPRi sensors, spectral SPRi sensors are more informative and have a wider detection dynamic range. According to the limited literature, spectral SPRi sensors include the following three types: (1) combining a fiber spectrometer with a mechanical scanner to achieve a two-dimensional (2D) distribution of RW for the sample[4,5]; (2) combining an wavelength scanner such as acoustic-optic tunable filter (AOTF) with a CMOS/CCD camera to obtain a series of wavelength resolved SPR greyscale images for making a RW map for the sample[6–8]; (3) recording SPR images with a RGB digital camera and then processing the images with the HSV color space model to achieve a 2D hue distribution for the sample[9,10]. The above-reported spectral SPRi sensors are insufficient in spectral resolution and/or spatial resolution.

Hyperspectral imaging emerges as an advanced spectral imaging technique with high spectral resolution and high image resolution, and it creates a hyperspectral datacube containing the spatially resolved spectral information of the sample and can provide both a greyscale image at every wavelength in the visible-near infrared (NIR) spectral region and a visible-NIR spectrum at each pixel of the image[11]. Hyperspectral imaging has been combined with other optical analyzing techniques to constitute a variety of powerful instruments, such as hyperspectral dark-filed microscope[12–14], hyperspectral fluorescence microscope[15], hyperspectral Raman microscope[16–18], and so on[19,20]. Hyperspectral imaging is also compatible with spectral SPRi sensors, and their combination will lead to a breakthrough in current SPR sensor technology.

In this work, we developed a hyperspectral surface plasmon resonance microscopy (HSPRM) system by optically connecting a prism-based SPR sensor to a hyperspectral microscope through an imaging lens and a mirror. The prepared HSPRM system can simultaneously provide full-color SPR images with pixels up to 1936 × 2202 per image, SPR greyscale images at the desired wavelengths in the spectral range of 400–1000 nm, and visible-NIR SPR spectrum at each pixel of the image with data points up to 1456 per spectrum (the wavelength separation of 0.41 nm). Every pixel of the HSPRM image can serve as an independent spectral SPR microsensor for quantifying microscopic objects. The HSPRM system utilizes a high-brightness broadband light source to ensure that the single-pixel spectral collection can be achieved within an exposure time as short as 0.5 ms and the imaging speed can reach 80 frames per second (fps). The lateral resolution of the prepared HSPRM system is up to 1.2 μm, and the system is equipped with three objectives of different magnifications to provide an optional FOV for SPR imaging. The HSPRM system is capable of measuring SPR spectra in the radiometric unit to improve the figure of merit (FOM) of single-pixel SPR sensors. Pixel-by-pixel calibration of the incident angle with the Fresnel model was carried out to eliminate sensitivity differences between different pixels due to slight beam divergence[21]. The 2D quantification capability of the prepared HSPRM system is fully demonstrated by experiments, and this capability enables our HSPRM system to outperform the existing SPR sensors. In this work, all SPR greyscale images obtained with the HSPRM system were enhanced using the 2D distribution of s-polarized intensity measured under the same conditions as a reference.

## Results

### Construction of HSPRM system

Our HSPRM system is schematically shown in Fig. 1a, and the corresponding lab-made actual system is shown in Fig. 1b. The HSPRM system contains two parts: a Kretschmann-type SPR sensor platform and an upright hyperspectral microscope, which are optically connected via an achromatic imaging lens and a mirror. The Kretschmann-type SPR sensor platform consists of a high-brightness broadband fiber-coupled lamp (LDLS EQ-99, Energetiq), a dispersion-free

reflective collimator (RC08SMA-P01, THORLABS), a Wollaston polarizer (FOCtek), and an isosceles right-angle glass prism. The hypotenuse length of the prism is 36.8 mm. The imaging lens (NA = 0.4, Daheng Optics, shown as L1 in Fig. 1a) is fixed at $d$ = 4 mm ~7 mm from the prism's refracting surface, and a SPR chip is attached to the base of the prism with an index-matching liquid. White light from the lamp is directed through a quartz fiber to the collimator to produce a collimated beam that becomes p-polarized after passing through the polarizer. The beam is then refracted into the prism at normal incidence and undergoes total internal reflection at the metal/glass interface of the SPR chip, resulting in evanescent excitation of surface plasmon wave (SPW) on the metal surface. The reflected and scattered light is refracted out of the prism and then collected by the imaging lens L1, resulting in a clear SPR image. With the mirror the SPR image plane is adjusted to be horizontal for easy observation with the upright hyperspectral microscope. The SPRi sensor equipped with the imaging lens L1 has a large FOV as indicated by the image P2 in Fig. 1a (FOV ~ 40 mm²). Without the imaging lens L1, only a blurred image (P1 in Fig. 1a) can be observed near the prism's refracting surface because the triangle prism does not affect the image stored in the reflected collimated beam but cannot converge the scattered light.

The upright hyperspectral microscope was assembled in our laboratory using a commercial hyperspectral imager (GaiaField Pro-V10E, Dualix Spectral Imaging) and a customized microscope, and it allows use as a standalone instrument. The hyperspectral imager has 1456 spectral channels and 4.26 megapixels and operates in push-broom mode[22], and it can provide both greyscale images with each spectral channel over the wavelength range from 400 nm to 1000 nm and visible-NIR spectrum of each pixel. The temporal resolution of the hyperspectral imager is limited by the push-broom mode, which can be improved by internal pixel fusion and spectral channel merging, and externally by using a high-power light source. In this work, the light source is a high-brightness LDLS EQ-99 lamp, which enables the hyperspectral imager to acquire a single-pixel spectrum in only 0.5 ms and to acquire images at an imaging speed of 80 fps. In this case, the prepared HSPRM system requires a minimum time of 3.4 s to acquire a hyperspectral SPR datacube with pixel fusion and spectral channel merging (datacube shown as P3). This optimal temporal resolution is a result of sacrificing the image resolution and spectral resolution. In experimental measurements with the HSPRM system, a trade-off is often made between temporal resolution, spectral resolution and image resolution. Notably, the hyperspectral imager used herein is capable of radiometric measurements of single-pixel spectra, and this unique capability can be used to improve the FOM of single-pixel spectral SPR sensors as demonstrated below. The customized microscope contains three different objectives (5x, 10x, 20x) to provide different FOVs and different image resolutions, and it is also equipped with a two-axis precision moving stage to shift the objective laterally to select the region of interest of the SPR image (P2) for hyperspectral microscopy. We do this because the SPR image (P2) is fixed during measurement. After the image area is selected, the hyperspectral observation is performed by adjusting the height of the objective to focus on the SPR image plane (P2). The resulting HSPRM system with the above configuration can be used either as a monochromatic SPRi sensor with flexible wavelength selection or as a spectral SPR microsensor or its high-density array. In some applications, SPR chips are covered with dielectric films, allowing not only SPR but also plasmon waveguide resonance (PWR)[23,24]. In this case, the prepared HSPRM system can simultaneously provide both spectral SPR and PWR images due to its broad operating wavelength range.

### FOV of the HSPRM system

The HSPRM system operates in a unique mode involving two steps: first generating a spectral SPR image of the sample using the achromatic imaging lens L1, and then analyzing the region of interest of this

image using the hyperspectral microscope. The HSPRM system can provide three types of SPR image: greyscale image, spectral image and RW image. Greyscale SPR images are the most common, and their quality depends on the lateral resolution of the HSPRM system, which is determined by the NA of the lens L1 and its distance from the refracting face of the prism. RW images are an exclusive feature of our HSPRM system and are superior to greyscale SPR images. This is so because RW relies on RI and/or thickness of the sample and is less affected by the scattered light and it can be used for the quantification of samples based on Fresnel theory. In addition, according to the principle of optical imaging, the FOV of the HSPRM system is dependent on the distance of the lens L1 from the prism's refracting face. It is

concluded that the longer the distance, the wider the FOV, but the lower the lateral resolution. To demonstrate this conclusion and the HSPRM performance, we fabricated a patterned SPR chip by standard photolithography using a 1.5-µm-thick positive photoresist layer (see Methods for chip fabrication). After characterizing the feature sizes of the patterned chip with an optical microscope (see Supplementary Fig. 1), we conducted SPR imaging of the chip using the HSPRM system. Fig. 2a and b each show three spectral SPR images obtained with 5x, 10x, 20x objectives. The distance from the lens L1 to the prism's refracting face is $d = 7$ mm in Fig. 2a and $d = 4$ mm in Fig. 2b. In each image, the exposed gold film area (green) and the photoresist-covered region (red) can be clearly distinguished with two different colors.

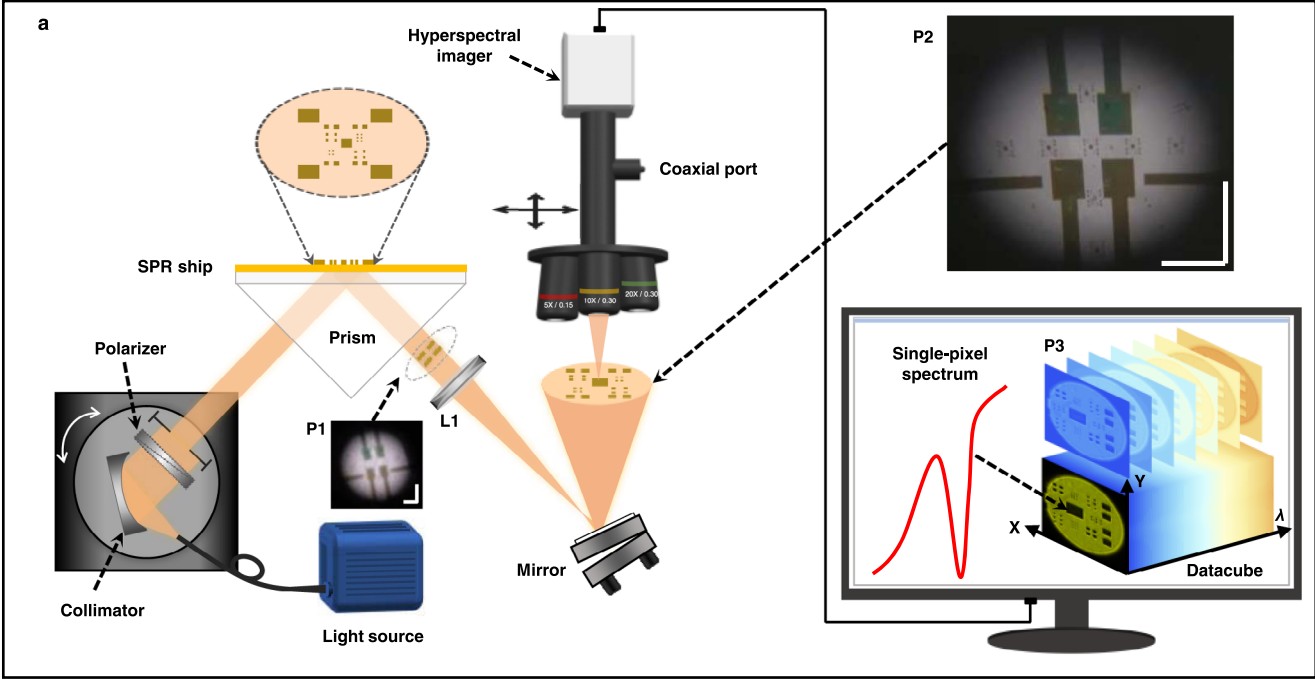

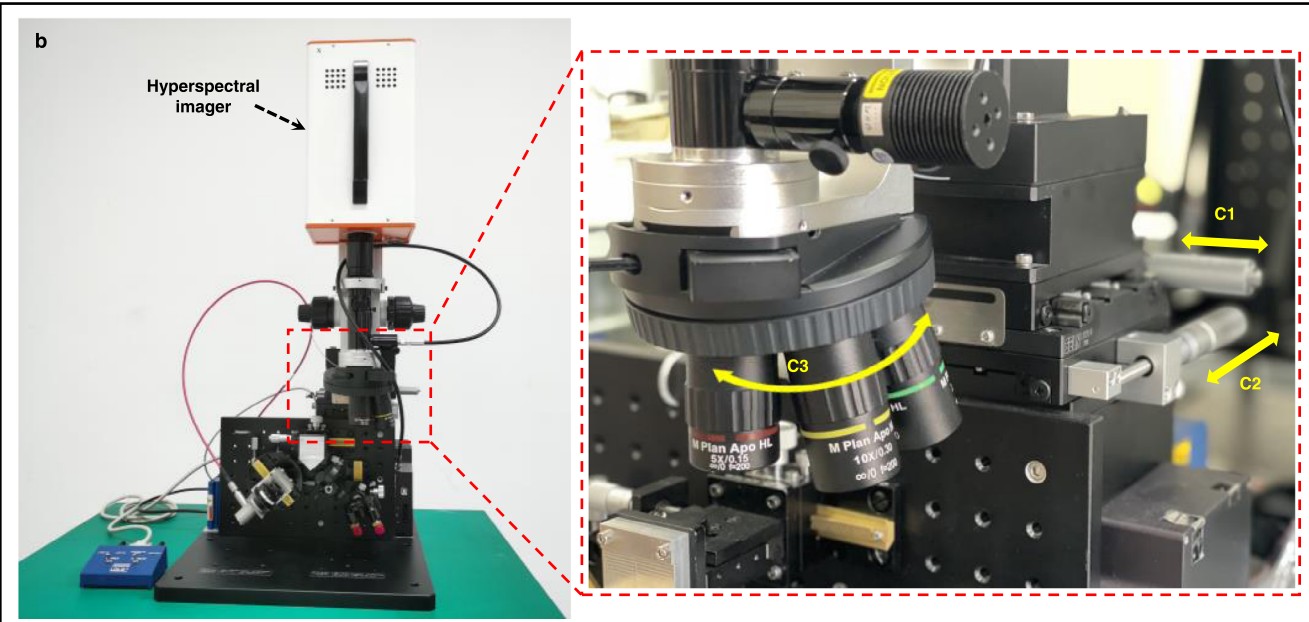

**Fig. 1 | HSPRM system constructed in our laboratory. a** Schematic diagram of the HSPRM system. P1 is a blurred SPR image contained in the reflected collimated beam. L1 is an achromatic imaging lens (NA = 0.4). P2 is a large FOV, clear SPR image formed by L1. P3 is a hyperspectral datacube for the selected area in

P2. Scale bar = 2 mm. **b** Photograph of the laboratory-made HSPRM system. C1 and C2 are x-axis and y-axis precision moving stages used for selecting the region of interest in P2. C3 is a rotating holder mounted with three objectives of different magnifications.

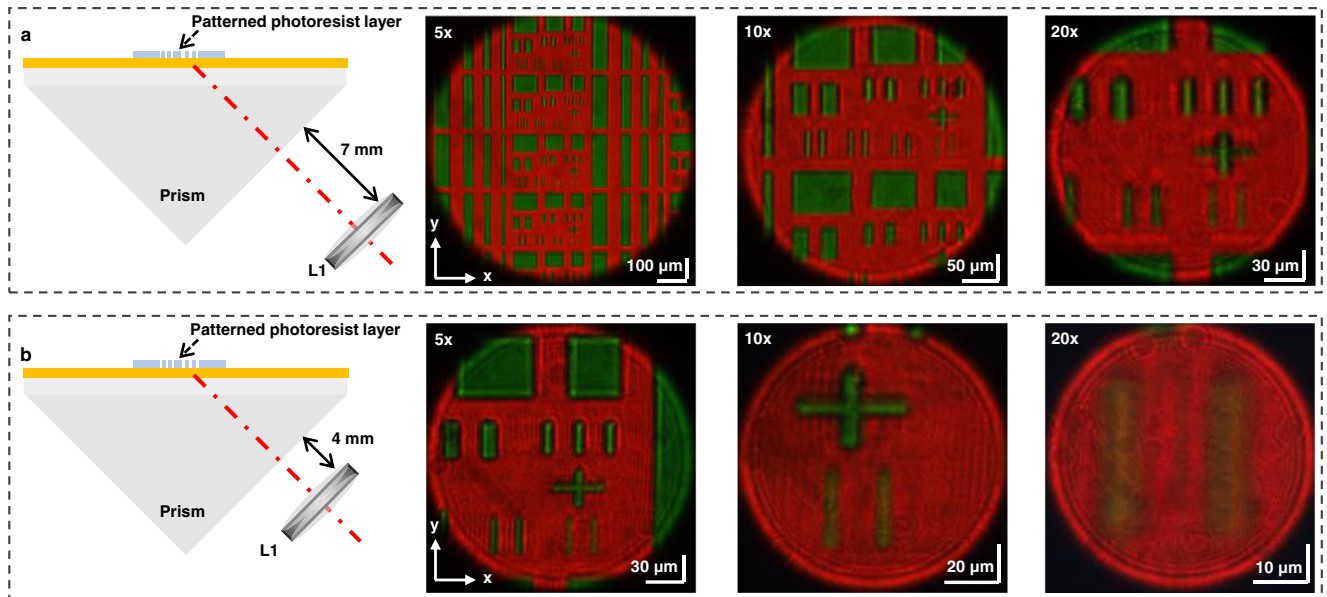

**Fig. 2 | Measurement of FOVs of the HSPRM system. a** Three spectral SPR images of a patterned SPR chip measured using the HSPRM system with a NA = 0.4 imaging lens fixed at $d = 7$ mm from the prism. The FOVs of three images corresponding to the 5x, 10x and 20x objectives are 0.884 mm², 0.145 mm² and 0.042 mm², respectively. **b** Similar results measured in the case of fixing the NA = 0.4 imaging lens at $d = 4$ mm from the prism. The FOVs of three SPR images are 0.049 mm², 0.010 mm², and 0.003 mm², respectively. In each image, the green areas are the exposed gold film and the red areas are the photoresist-covered gold film. At least three measurements were performed on different samples with similar results.

These two colors result from SPR on the exposed gold film surface and PWR in the film-covered region. The FOVs of three images in Fig. 2a are 0.884 mm², 0.145 mm² and 0.042 mm², much larger than those of the corresponding images in Fig. 2b (0.049 mm², 0.010 mm² and 0.003 mm²), and the minimum FOV in Fig. 2a is very close to the maximum FOV in Fig. 2b. The combination of Fig. 2a and b indicate that our HSPRM system can easily provide spectral SPR images with different FOVs by using objectives of different magnifications. A comparison of Fig. 2a and b reveals that the FOV of the HSPRM system can also be effectively adjusted by changing the distance of the lens L1 from the prism's refracting face. In all the above spectral SPR images, the y-axis is the SPW propagation direction and the image size is compressed along the y-axis direction due to the oblique illumination of the SPR chips[25]. It is worth noting that the quality of the spectral SPR image of two 1.6 μm-wide parallel bars obtained with the 20x objective is degraded by the insufficient light intensity for the given exposure time.

**Greyscale images and RW images of the patterned SPR chip**
Each spectral SPR image in Fig. 2 is a hyperspectral datacube. We can extract the greyscale SPR image at the desired wavelength from the measured hyperspectral datacube, and we can also achieve the RW image by determining the RW at each pixel from a huge number of single-pixel SPR spectra. Fig. 3a and b show two greyscale SPR images at the wavelength of 515 nm, extracted from the 20x spectral image in Fig. 2a and the 5x spectral image in Fig. 2b, respectively. We select these two images for comparison because their FOVs are close to each other. The greyscale image in Fig. 3b is clearer than that in Fig. 3a, as seen with the naked eye. The dark-field background and bright-field patterns for the two images indicate that at the wavelength of 515 nm, PWR appears in the photoresist-covered area, while SPR disappears from the exposed gold film surface. The other two plots in Fig. 3a show the normalized intensity distributions along the x-axis and y-axis for the three regions marked A, B, and C in the greyscale image, respectively. The three regions marked correspond to the exposed gold film areas with the equal actual sizes of 5 μm × 35 μm. As can be seen from the two normalized intensity profiles, the dimensions of the three marked regions are determined as: 8 μm × 46 μm for region A, 7 μm ×

45 μm for region B and 8 μm × 45 μm for region C, all larger than the actual size in the x-axis and y-axis directions. The blurred edges of each marked region reveal the insufficient lateral resolution of the lens L1. Fig. 3b also includes the two normalized intensity distribution plots for the same regions as those in Fig. 3a. From the two plots, the dimensions of the three regions marked are obtained as 7 μm × 42 μm for region A, 7 μm × 41 μm for region B, and 8 μm × 41 μm for region C. The size of each region obtained in Fig. 3b is larger than its actual size but smaller than the corresponding value obtained in Fig. 3a. This comparison combined with the result in Fig. 2 verified that increasing the distance of the lens L1 from the prism's refracting face can result in an increased FOV but a reduced lateral resolution. Due to the contribution of SPW propagation length, the difference between the measured and actual size of the region is larger in the y-axis direction ($\Delta L_y \geq 6$ μm) than in the x-axis direction ($\Delta L_x \leq 3$ μm). In addition, $\Delta L_y$ in Fig. 3a is larger than that in Fig. 3b, suggesting that the influence of the SPW propagation length on the SPR image can be weakened by increasing the lateral resolution of the HSPRM system. The above experimental results demonstrated that for the prism-based SPR imaging with the collimated incident beam, the efficient collection of forward scattered light is essential for improving SPR greyscale image quality, especially for sharpening the edges of greyscale images. In this work, the lens L1 is typically mounted $d = 4$ mm from the refracting face of the prism for routine high-resolution spectral SPR imaging measurements.

Fig. 3c, d display two RW images, corresponding to the 20x spectral image in Fig. 2a and the 5x spectral image in Fig. 2b, respectively. Each RW image is a 2D distribution of single-pixel-resolved RWs for the SPR mode ($\lambda_R = $~592 nm) in the exposed gold film areas and for the PWR mode ($\lambda_R = $~515 nm) in the photoresist-covered areas. Fig. 3c, d present the similar geometric patterns as those in Fig. 3a, b, confirming that the RW image can reflect the surface patterns of the SPR chip at least as reliably as the SPR greyscale image. For detailed comparison with the SPR greyscale images, the two RW distributions along the x-axis and y-axis were obtained from the corresponding RW images, as shown in Fig. 3c, d. All the RW distribution plots are assigned to the exposed gold film areas with the equal sizes of 5 μm × 35 μm. These RW profiles exhibit abrupt changes without gradients at the edges of

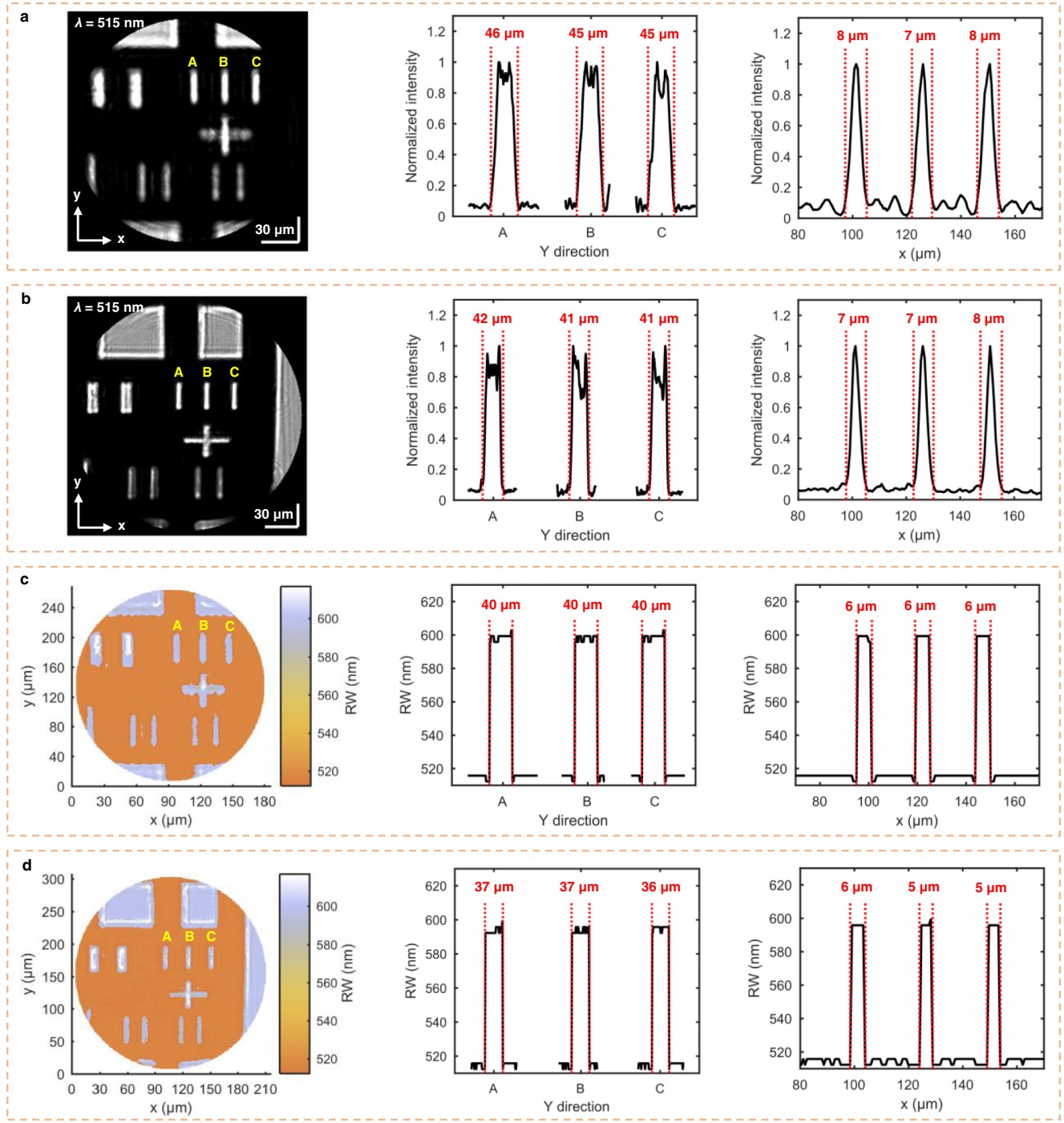

**Fig. 3 | Greyscale images and RW images of the patterned SPR chip. a** Greyscale image at 515 nm wavelength and normalized intensity distributions along the x- and y-axes for the regions labeled A, B and C in the image. The greyscale image was extracted from the hyperspectral datacube acquired with the 20x objective and the NA = 0.4 imaging lens fixed at $d$ = 7 mm from the prism. **b** Similar results obtained with the 5x objective and fixing the NA = 0.4 imaging lens at $d$ = 4 mm from the prism. **c** and **d** RW images and RW distributions along the x-axis and y-axis for the regions labeled A, B and C in each image. The RW images in **c** and **d** correspond to the greyscale images in **a** and **b**, respectively. The regions labeled A, B, and C in the images have the actual size of 5 μm × 35 μm.

the marked regions, much better than the corresponding normalized intensity profiles in Fig. 3a, b. The results show that the RW image is not sensitive to scattered light and thus more resistant to noise than the greyscale SPR image. Using the 1D profiles of RWs in Fig. 3c, the regions marked A, B, and C are determined to be 6 μm × 40 μm, respectively, larger than the actual size but better than the results obtained using the normalized intensity profiles in Fig. 3a. More notably, the sizes of the marked regions obtained using the RW profiles in Fig. 3d are closer to the actual values than those in Fig. 3c. The comparison shows that

the RW image is superior to the greyscale SPR image in terms of image quality and dimensional accuracy.

## Lateral resolution measurement

As demonstrated above, our HSPRM system can provide both grey-scale image at the desired wavelength and high-quality RW image. The RW-based SPR imaging is currently in its early stage, and there is a lack of reports on the RW-based lateral resolution and its measurement method. Herein we investigate the lateral resolution of the

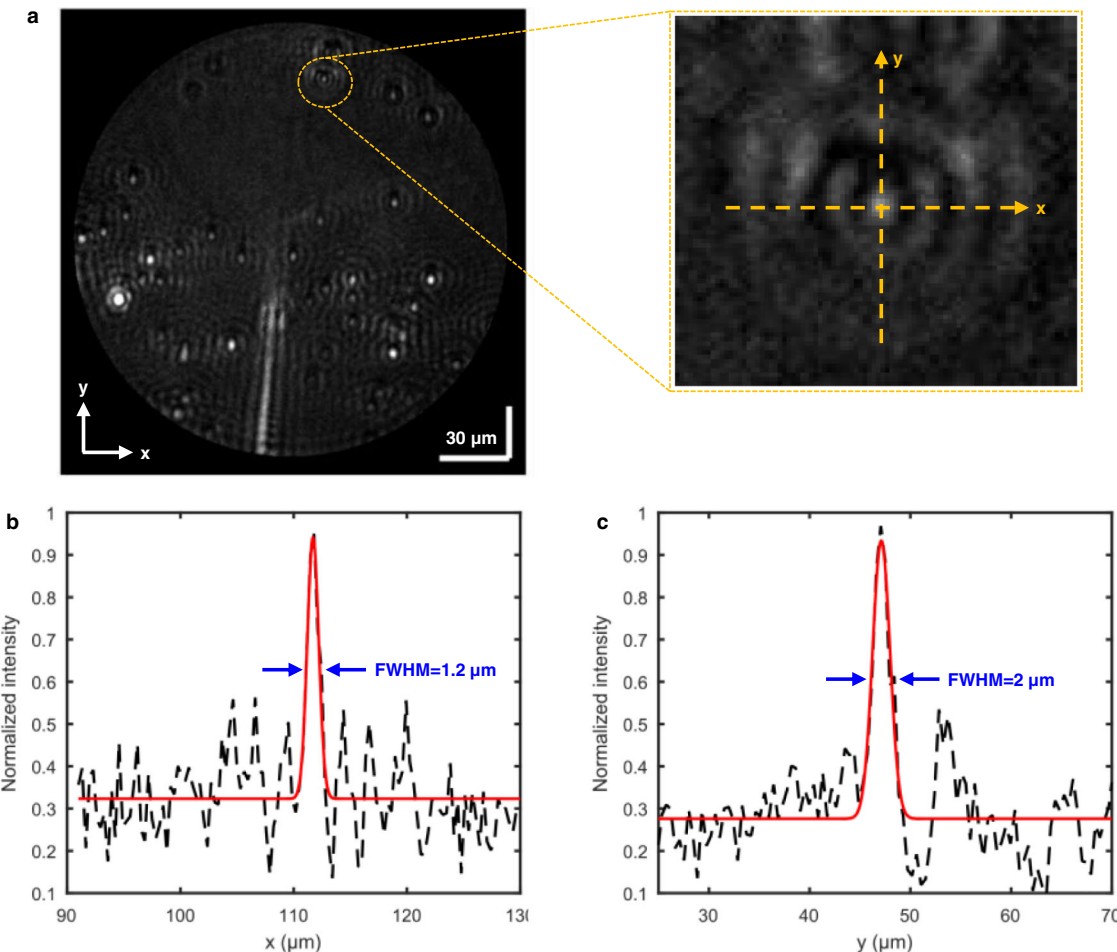

**Fig. 4 | Measurement of spatial resolution of the HSPRM system. a** SPR greyscale image at $\lambda = 592$ nm for the PS nanospheres attached to the gold-film SPR chip (inset: an enlarged view of the selected area, and the x and y directions indicate the directions perpendicular to and parallel to the SPW vector). **b** and **c** Measured lateral resolutions in the x and y directions. The black dashed curves are the experimental PSF and the red solid curves are Gaussian fitting curves to the experimental data. The FWHM referred to as the spatial resolution is 1.2 μm and 2 μm in the x and y directions, respectively. The similar results were obtained using two different gold-film SPR chips. The colloidal solution of PS spheres with size distribution from 0.6 μm to 1.0 μm were purchased from Shanghai Aladdin.

HSPRM system by measuring the point spread function (PSF) of the system using the greyscale SPR image of a point object. The PSF acquisition requires the size of the point objects to be smaller than the spatial resolution of the system[26]. The existing prism-based SPRi sensors generally have a spatial resolution of ca. 3 μm[27]. Therefore, we used polystyrene (PS) spheres of less than 1 μm in diameter as the point objects for SPR imaging. We purchased the PS sphere colloidal solution (0.6–1.0 μm) from Shanghai Aladdin and then immobilized PS spheres on a gold-film SPR chip. After dried at room temperature, the PS spheres on the SPR chip were imaged using the HSPRM system with the 5x objective. Fig. 4a shows the SPR greyscale image at $\lambda = 592$ nm, extracted from the measured hyperspectral SPR data-cube. The dark background of the image indicates that SPR occurs on the surface of the gold film at $\lambda = 592$ nm. The image contains some bright spots surrounded by bright rings, representing the isolated PS spheres and their Airy disk pattern. We select a smaller spot in the image to acquire the PSF. The inset shows an enlarged view of the selected area, where the y-axis represents the SPW propagation direction. Fig. 4b, c show the normalized intensity distributions along the x-axis and y-axis for the selected area and their best-fitting curves with Gaussian function. The full width at half maximum (FWHM) of the intensity peak in the best-fitting curve represents the lateral resolution[28]. Using the experimental data in Fig. 4b, c, the lateral resolution of our HSPRM system is determined to be 1.2 μm and 2 μm in the directions perpendicular and parallel to the SPW vector,

respectively. This spatial resolution is better than most existing prism-based SPRi sensors[8,26,27,29]. Our experimental results indicate that the lateral resolution obtained with the 5x objective is better than that with the 20x objective. This is because the lateral resolution is mainly dominated by the lens L1 not the objective. The second reason is that the SPR greyscale image obtained with the 20x objective has a lower signal-to-noise ratio due to the limited luminous flux, resulting in a drop in the lateral resolution determined from the measured intensity profile.

## Single-pixel spectral SPR sensing

Single-pixel spectral SPR sensing is one of the fundamental capabilities of our HSPRM system, and different pixels must have the same SPR sensitivity to liquid RI for accurate 2D quantification of solid-state chemical and biological measurands. To investigate RI sensitivities of different pixels of the HSPRM system, we prepared a series of aqueous glycerol solutions with different RIs ranging from 1.3266 to 1.3491, and then we measured the hyperspectral SPR images of these RI solutions by loading each solution sample onto the gold-film SPR chip and fixing the incident angle at $\theta \approx 40°$. During the measurement, the lens L1 was not used and only the 5x objective was used to collect as much reflected light as possible (FOV ~ 4 mm²) for the purpose of evaluating the divergence of the incident collimated beam. We do this because the beam divergence can cause considerable differences in RI sensitivity between different pixels. Spectral SPR images measured at a

series of glycerol solutions are provided in Supplementary Fig. 2, and Fig. 5a, b show the selected two spectral SPR images with maximum color difference, corresponding to the glycerol solutions with RI = 1.3266 and RI = 1.3491, respectively. Each image represents a hyperspectral datacube from which a 2D distribution of RW can be obtained. Fig. 5c displays a series of 2D distributions of RW, each of which corresponds to a glycerol solution with different RI. It is emphasized here that accurate RW acquisition was achieved by converting the single-pixel SPR intensity spectra into the corresponding radiance spectra using the radiometric correction algorithm included in the hyperspectral imager software. Radiometric correction is an essential image-processing function of a hyperspectral imager for remote sensing[30,31], which involves subtracting the background signal (bias) and dividing by the gain of the instrument and consequently converting the raw instrument output (digital numbers) to radiance. For comparison, we provided the RI-dependent SPR intensity spectra (Fig. 5d) and the corresponding radiance spectra (Fig. 5e) for the pixel marked C in Fig. 5c. Evidently, the resonance valleys in the SPR radiance spectra are much better in shape than those in the original SPR intensity spectra, meaning that RW can be more accurately determined from the SPR radiance spectrum than from the SPR intensity spectrum. Moreover, the SPR radiance spectrum can effectively enhance the FOM of the single-pixel spectral SPR sensor. The FOM is an important quantity used to evaluate the performance of a SPR sensor, which is defined as a ratio of the sensor's RI sensitivity to the FWHM of the SPR resonance valley[32]. As shown below, the RI sensitivity of single-pixel spectral SPR sensor is typically 3050 nm/RIU. Therefore, with the SPR radiance spectrum the FOM is obtained to be 44.8 RIU$^{-1}$, larger than that obtained with the SPR intensity spectrum (FOM = 31 RIU$^{-1}$), as shown in Fig. 5f.

Using the RI-dependent RW data in Fig. 5c, the relationship between the RW at a given pixel and the solution RI was obtained to be linear, with its slope representing the RI sensitivity at that pixel. Plotting RI sensitivities against pixels leads to a 2D sensitivity distribution, as shown in Fig. 6a. It can be seen from Fig. 6a that there exist sensitivity differences between different pixels in the image FOV. To analyze the directionality of the sensitivity divergence, the sensitivity distributions along x-axis and y-axis are shown in Fig. 6b (the y-axis is the SPW propagation direction). Obviously, the RI sensitivity almost linearly decreases with increasing pixels along the y-axis. In contrast, the RI sensitivity along the x-axis remains stable within the limited range of fluctuations. The comparison between the sensitivity distributions along the x-axis and the y-axis reveals that the incident collimated beam remains slightly divergent. The beam divergence should be symmetric about the x-axis and y-axis, however, only the beam divergence along the y-axis affects the angle of incidence and thus the RI sensitivity. The calibrated angle of incidence at each pixel of the SPR image was obtained by fitting the measured single-pixel RI sensitivity using the three-layer Fresnel formula (see Methods and Supplementary Fig. 3). As shown in Fig. 6c, the calibrated incident angle increases quasi-linearly from 39.80° to 40.07° as the number of pixels along the y-axis increases. This means that the incident collimated beam impinging on the image FOV has a divergence angle of 0.27°.

Using the pixel-by-pixel calibrated incident angles, the HSPRM system enables accurate 2D quantification of nanometric measurands. To demonstrate this capability, a large-area uniform ultrathin film of titanium dioxide (TiO$_2$) was sputtered on the gold film SPR chip for imaging analysis using the HSPRM system. Fig. 6d shows the spectral SPR image of the TiO$_2$ film in air obtained at normal incidence with the 5x objective but without the lens L1, and the image FOV is ca. 2 mm$^2$. The 2D distribution of RWs was derived from the measured hyperspectral datacube, which was then fitted using the four-layer Fresnel formula (see Methods) and the calibrated incident angles. As the fitting result, a 2D thickness distribution of

the TiO$_2$ film is shown in Fig. 6f. It can be seen from Fig. 6f that the film thickness is flatly distributed over the entire FOV with an average thickness of 3.5 nm and a fluctuation of ±0.1 nm. For comparison, the fitting calculation was also performed in the same way but without the angular calibration. Fig. 6e shows the 2D thickness distribution of the TiO$_2$ film obtained without the angular calculation. In this case, the film thickness exhibits a tapered distribution along the y-axis with a minimum thickness of 3.3 nm ± 0.1 nm and a maximum thickness of 3.6 nm ± 0.1 nm in the image FOV. Such tapered thickness distribution is related to the quasilinear divergence of the incident collimated beam. The standard deviation of the film thickness is 0.1 nm before the angular calibration and 0.05 nm after the angular calibration. The measurement precision increases 2 times with the angular calibration. The above-measured thickness of the TiO$_2$ film was verified with an ellipsometer, which gave a thickness of 3.77 nm for a 1 cm diameter area of the TiO$_2$ film (see Supplementary Table 1). It is worth noting that the standard deviation of film thickness due to the incident beam divergence is highly dependent on the film RI, and the lower the film RI, the larger the standard deviation. We simulated the thickness distribution along the y-axis of an adsorbed bovine serum album (BSA) layer (thickness = 3.5 nm and RI = 1.429) using the above-calibrated incident angles (Fig. 6c). The simulation results show a tapered distribution of BSA film thickness along the y-axis with a minimum thickness of 2.8 nm and a maximum thickness of 4.3 nm in a ~2 mm$^2$ FOV (provided in Supplementary Fig. 4). The simulated standard deviation of the BSA adlayer is 0.36 nm, which is 3.6 times as large as that of the 3.5-nm-thick TiO$_2$ film measured above, evidencing that the standard deviation of film thickness increases with decreasing the film RI. Therefore, it is concluded that even a small divergence of the incident angle can seriously affect the quantified results of biochemical samples and thus the angular calibration is very important.

## Measurement of a monolayer graphene

To demonstrate the spatially resolved detection and quantification capabilities of the HSPRM system, a 2D nanomaterial sample was prepared by transferring a monolayer graphene onto the gold film of the SPR chip using a wet transfer method (see Methods). Spectral image of the sample in air was obtained using the HSPRM system at normal incidence and with the lens L1 and 5x objective combination. As shown in Fig. 7a, the graphene-covered area and the exposed gold-film area show two different colors, and the boundary between the two areas is distinguishable. Fig. 7b displays a 2D distribution of RWs of the sample, showing that the single-pixel RW in the graphene-covered area are larger than those in the exposed area. The incident angle calibration was performed using the single-pixel RW in the exposed area (provided in Supplementary Fig. 5), and then the 2D thickness profile of the monolayer graphene was reconstructed from the 2D distribution of RWs in the graphene-covered area. The reconstruction calculation was carried out using the four-layer Fresnel formula, the calibrated incident angles and the given graphene RI 2.74 + 1.47i[33]. Fig. 7c shows the reconstructed thickness distribution of monolayer graphene, and based on that, the single-pixel thickness profile (Fig. 7d) and the statistical distribution (Fig. 7e) of thicknesses at about 120,000 pixels in the graphene-covered area are obtained. Fig. 7d, e reveal that the measured monolayer graphene thickness is mainly distributed between 0.5 nm and 1 nm, and the average thickness is determined to be 0.71 nm, very close to the reported value of 0.82 nm measured by the SPR method[34]. Fig. 7f shows an optical microscope image of the sample, on which the exposed and graphene-covered areas can also be distinguished.

## Detection of protein adsorption on an inhomogeneous film

In addition to the aforementioned ability to quantify ultrathin films, our HSPRM system can also be used for detection and quantification of

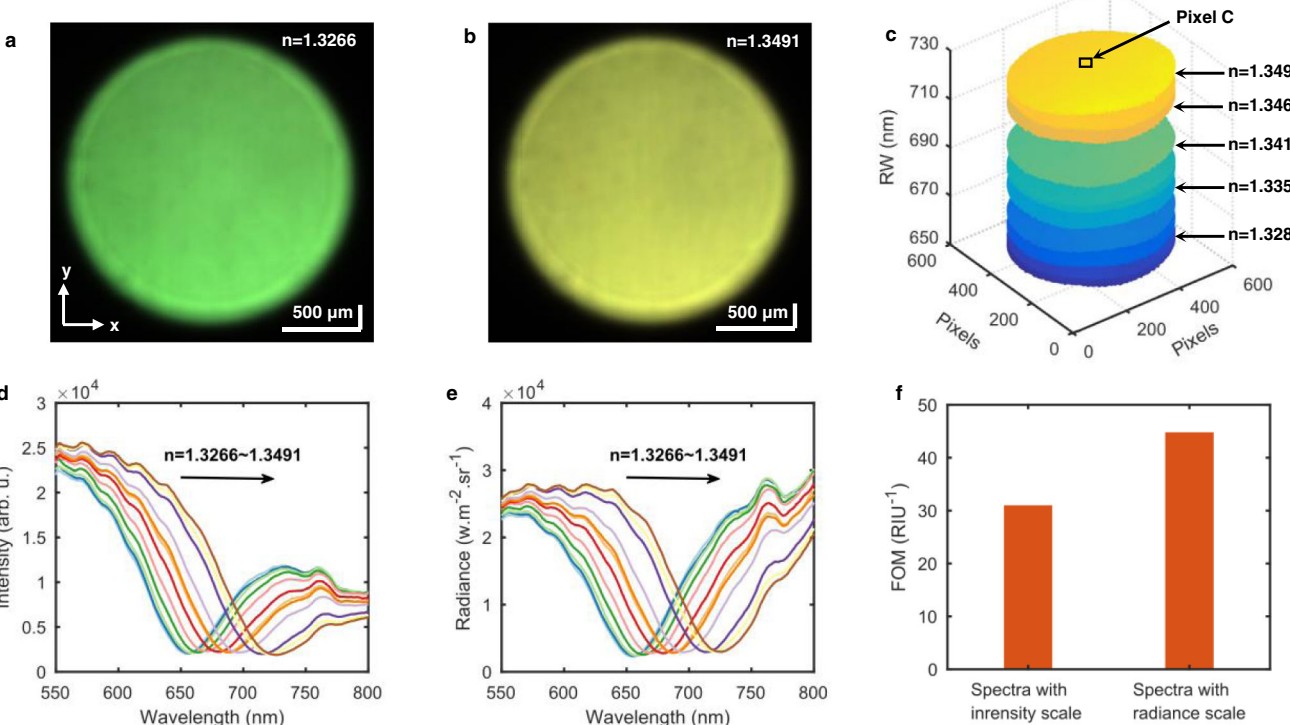

**Fig. 5 | Responses of the HSPRM system to changes in bulk RI. a** and **b** Spectral SPR images of the gold film covered with aqueous glycerol solutions of RI = 1.3266 and RI = 1.3491. **c** RW images determined from the hyperspectral SPR datacubes measured with a series of glycerol aqueous solutions with RI ranging from 1.3266 to 1.3491. The pixel marked C was selected for single-pixel spectral analysis. **d** and **e** Single-pixel SPR intensity spectra and corresponding SPR radiance spectra measured with a series of glycerol aqueous solutions with RI ranging from 1.3266 to 1.3491. As RI increases, the resonance dip shifts towards longer wavelengths. **f** Comparison of the FOM determined from SPR intensity spectrum and SPR radiance spectrum. The similar results were obtained with three repeated measurements.

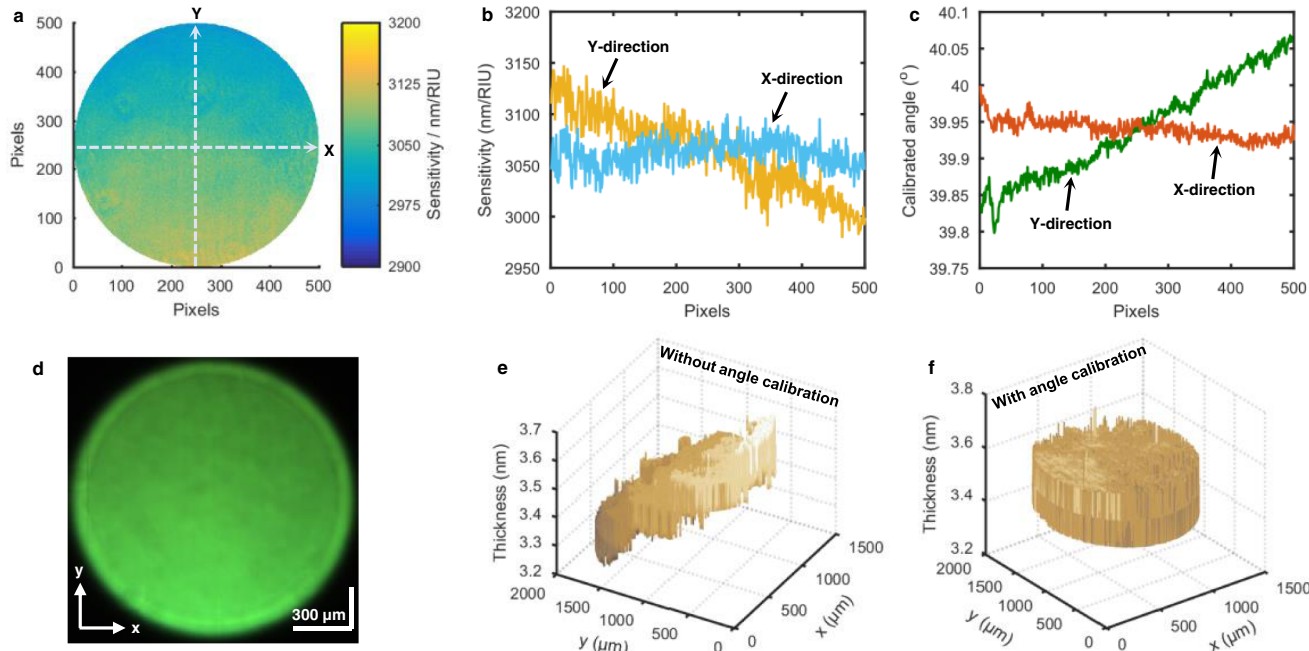

**Fig. 6 | Pixel-by-pixel calibration of incident angle and calibration effect test. a** 2D distribution of RI sensitivity determined from the measured RW data in Fig. 5c. The x and y directions are perpendicular to and parallel to the SPW vector. **b** RI sensitivity at different pixels along the dotted lines in Fig. 6a. **c** Calibrated incident angles at different pixels along the dotted lines in Fig. 6a. **d** Spectral SPR image of the TiO₂ ultrathin film in air. **e** and **f** 2D thickness distributions of the TiO₂ ultrathin film calculated without and with calibration of incident angle. The similar results were obtained with four repeated measurements.

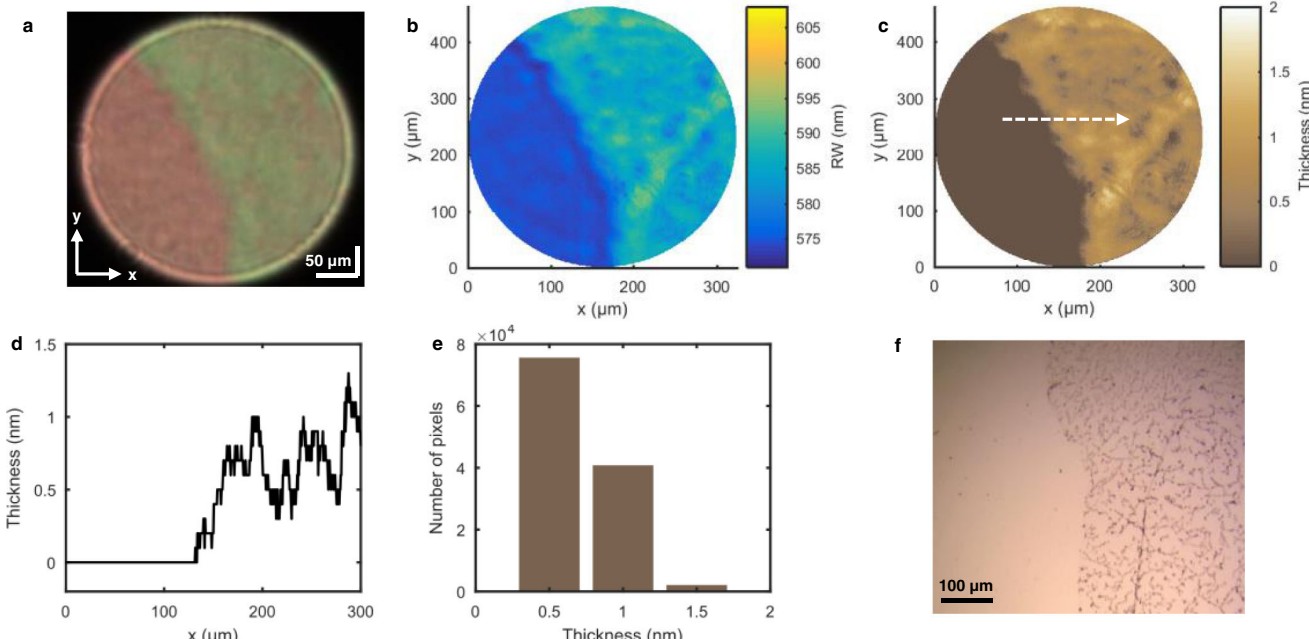

**Fig. 7 | Quantification of monolayer graphene thickness using the HSPRM system. a** Spectral SPR image of the monolayer graphene covered SPR chip. **b** RW image extracted from the measured hyperspectral datacube. **c** Reconstructed 2D thickness distribution of the monolayer graphene. The arrow line indicates the path to analyze the thickness fluctuation. **d** Thickness fluctuation along the arrow line in Fig. 7c. **e** Statistical distribution of thicknesses at about 120,000 pixels in the graphene-covered area in Fig. 7c. **f** Optical microscope image of the same graphene sample. The similar results were obtained with two samples.

biomolecular adsorption. Herein we use the HSPRM system to investigate BSA protein adsorption on a microstructured poly (methyl methacrylate) (PMMA) thin film coated on the gold-film SPR chip (see Methods for preparation of the PMMA film). Fig. 8a shows a dark-field optical microscope image of the as-prepared PMMA film, from which the wrinkled microstructure can been seen. The spectral SPR image of the PMMA film covered with deionized water was measured using the HSPRM system at the incident angle of $\theta = 20°$ and with the lens L1 and 5x objective combination. As shown in Fig. 8b, a wrinkled microstructure similar to that in Fig. 8a also appears on the spectral SPR image, indicating that the thickness of the PMMA film is smaller than the plasmonic field penetration depth. Fig. 8c displays the 2D distribution of RWs derived from the hyperspectral SPR datacube. Fig. 8d shows the 2D thickness distribution of the PMMA film reconstructed from the 2D RW distribution based on the four-layer Fresnel model. Fig. 8e presents the single-pixel thickness profile along the dotted line in Fig. 8d, from which the groove depth was determined to be larger than 5 nm.

Protein adsorption on the PMMA thin film was achieved after changing the medium overlaid on the thin film to a solution of 50 μM BSA in deionized water. The spectral SPR image of the BSA-adsorbed PMMA thin film was measured under the same conditions as mentioned above, which was then used to determine the 2D RW distribution. A pixel-by-pixel comparison between the 2D RW distributions obtained before and after BSA adsorption results in a 2D distribution of RW difference (ΔRW), as shown in Fig. 8f. ΔRWs are different at different pixels, and a large ΔRW means a large amount of adsorbed BSA molecules. Therefore, Fig. 8f reveals the inhomogeneous adsorption of BSA molecules on the wrinkled PMMA thin film. Note that ΔRW mainly results from the adsorbed BSA molecules and the contribution of the RI difference between the BSA solution and deionized water is too small to consider (see Supplementary Fig. 6).

On the basis of the five-layer Fresnel model shown in Fig. 8i, the BSA adlayer thickness at a pixel on the spectral SPR image in Fig. 8b was calculated using the RW measured at that pixel. The resulting 2D thickness distribution of the BSA adlayer was reconstructed, as shown

in Fig. 8g. Fig. 8h displays the thickness distribution of the BSA adlayer along the dotted line in Fig. 8g. A comparison between Fig. 8h and e indicates that the BSA adlayer thickness is larger at the bottom of the grooves of the PMMA thin film than at the top of the grooves, indicating that the amount of adsorbed BSA molecules is larger at the bottom of the groove than at its top. The above experimental results verify that our HSPRM system is indeed capable for spatially resolved quantification of biomolecular adsorption.

## Quantification of onion cell wall

Measurement of the optical parameters of onion tissues is an effective method for onion quality inspection[35]. We peeled a monolayer of onion epidermal cell walls from an onion and attached it to the gold film of the SPR chip for use as a biosample. Fig. 9a shows a dark-field optical microscope image of the biosample, and the onion epidermal cell walls are clearly seen in the image. We also investigated the biosample using the HSPRM system. Fig. 9b shows the spectral SPR image of the biosample in air measured at normal incidence and with the lens L1 and 5x objective combination. The observed area of the biosample in the spectral SPR image is the same as the dark-field microscope image, so the contours of onion epidermal cell walls in these two different images are identical to each other. The yellowish and green regions in the spectral SPR image, corresponding to the dark and light regions in the dark-filed microscope image, represent the cell wall attached area (without air gap) and cell wall non-attached area (with air gap). We selected two different pixels, A and B, in the SPR image to analyze their resonance spectra. Fig. 9c shows the spectrum of pixel A selected in the cell wall non-attached area, which contains a single resonance valley, corresponding to the SPR mode. Fig. 9d shows the spectrum of pixel B selected in the cell wall attached area, which contains four well-defined resonance valleys, indicating that the cell wall forms the PWR structure together with the gold film. Onion cell wall is mainly composed of cross-linked polysaccharides including cellulose, hemicellulose, and pectin (RI is about 1.5), and the cell wall thickness is in the micrometer order[36], thus the cell wall can support PWR mode. Note that the spectra in Fig. 9c, d were obtained by

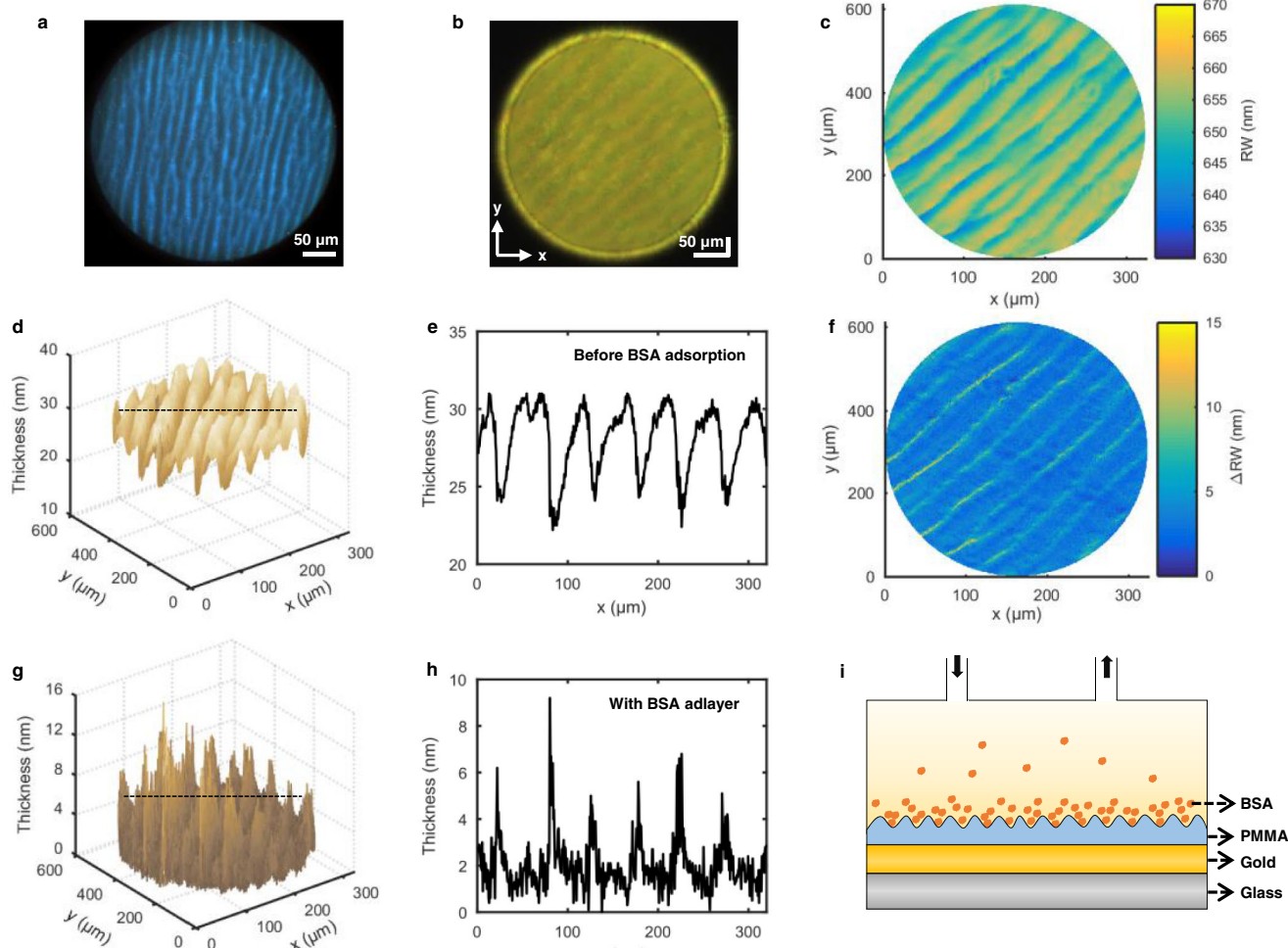

**Fig. 8 | Experimental results of protein adsorption on the wrinkled PMMA film.** **a** Surface morphology of the wrinkled PMMA film observed by a dark-field optical microscope. **b** Spectral SPR image of the wrinkled PMMA film before BSA adsorption. **c** RW image extracted from the hyperspectral SPR datacube acquired before BSA adsorption. **d** 2D thickness distribution of the wrinkled PMMA film. **e** Thickness distribution along the dotted line in Fig. 8d, showing the valley depth ≥ 5 nm. **f** 2D distribution of RW difference (ΔRW). ΔRW at a pixel is obtained by subtracting the RW before BSA adsorption from the RW after BSA adsorption at the same pixel. **g** 2D distribution of the BSA adlayer thickness. **h** BSA adlayer thickness fluctuation along the dotted line in Fig. 8g. The peaks of adlayer thickness correspond to the dips of PMMA film thickness in Fig. 8e. **i** Five-layer Fresnel model used for calculating BSA adlayer thickness. The similar results were obtained with two PMMA-coated SPR chips.

dividing the measured radiance spectra by the radiance spectra of the light source. The PWR spectrum of pixel B was fitted by using the spectrum simulation based on the four-layer Fresnel model (see Methods), resulting in the cell wall thickness of 1.973 μm and its effective RI of 1.542 at pixel B. The obtained cell wall thickness is close to the reported value of 1.62 ± 0.36 μm[36]. Furthermore, we derived the 2D RW distribution from the measured hyperspectral SPR datacube of the biosample, and then obtain the 2D thickness and effective RI distributions of the cell walls, as shown in Fig. 9e, f. The above experimental results show that our HSPRM system is very efficient and informative for cell analysis, and it can clearly distinguish the degree of adhesion of cells to the substrate, and its 2D quantification capability makes it superior to existing SPRi sensors.

## Discussion

We have developed a HSPRM system with outstanding capability for pixel-by-pixel accurate quantification of chemical and biological measurands over a large dynamic detection range. The HSPRM system consists of a prism-based spectral SPR sensing platform and a hyperspectral microscope, which are optically connected by an imaging lens and a mirror. The HSPRM system utilizes the imaging

lens to collect both scattered and reflected light from the SPR chip to form a SPR image with a large FOV, and then use the hyperspectral microscope to magnify the selected region of interest in the SPR image for spatially resolved quantitative analysis. The scattered light results from the out-of-plane scattering of the SPW at the edges of the object under study[37], which means that at the scattering points at the object's edges, SPW is directly converted into photons radiated from the SPW propagation plane. Only a part of the photons is refracted out of the prism along with the reflected beam and collected by the imaging lens. The collected photons are beneficial for improving the clarity of SPR image of the object because the scattering points at the object's edges can be traced back along their propagation paths. On the contrary, the in-plane scattering of SPW produces the scattered SPWs, which have the equal propagation constant but different propagation direction. The scattered SPWs propagate away from the scattering points before being coupled into directional light radiation by the prism. The collection of these directional light waves can result in blurred image edges, degrading SPR image quality. The triangle prism can output the SPR image contained in the reflected collimated beam, but the edges of the SPR image are blurred due to the inability of the prism to converge the

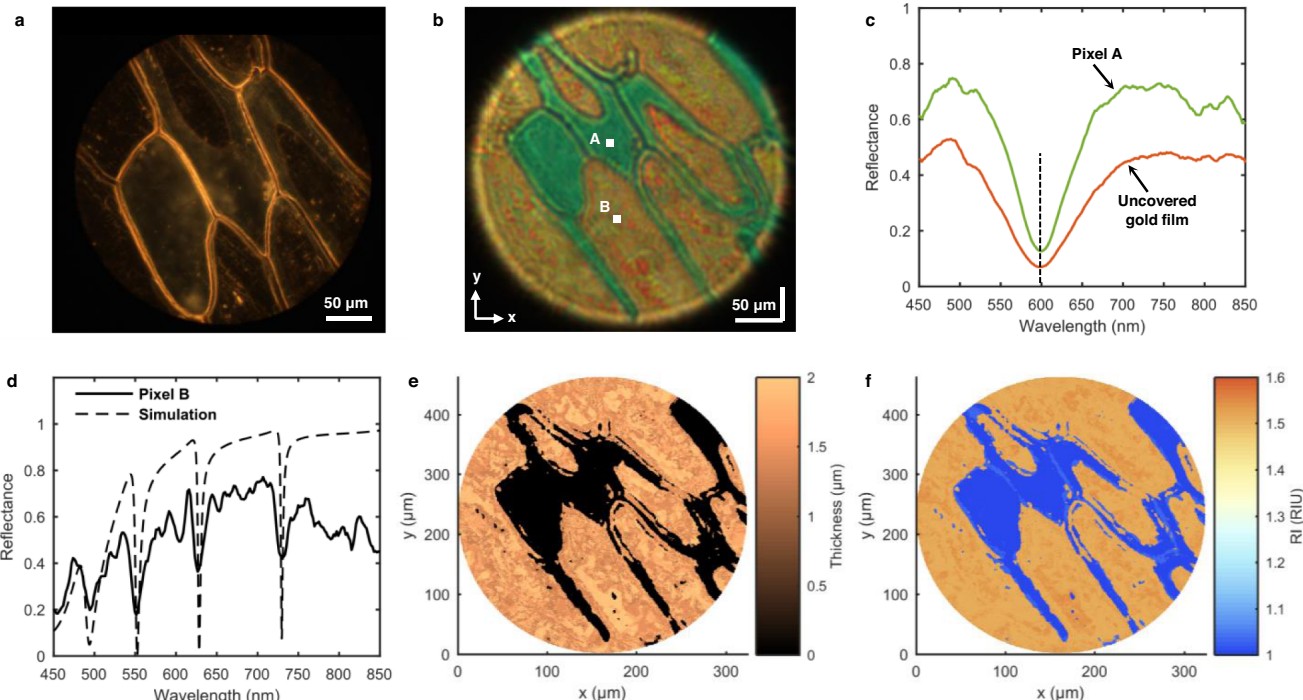

**Fig. 9 | Label-free measurement of onion epidermal cells. a** Surface morphology of onion monolayer epidermal cell wall covered on the gold-film SPR chip observed by the dark-field microscope. **b** Spectral SPR image of the same onion cell area as in Fig. 9a. **c** Single-pixel SPR spectrum of the exposed gold film (without coverage) and the SPR spectrum at the pixel marked A in the cell wall covered area in Fig. 9b. **d** Measured PWR spectrum at the pixel marked B in the cell wall covered area in Fig. 9b. The simulated spectrum is also shown for quantifying the cell wall thickness and RI. **e** 2D thickness distribution of the cell walls attached to the substrate. **f** 2D effective RI distribution of the cell walls attached to the substrate. The similar results were obtained with two independent measurements.

scattered light. The clarity of the SPR image can be improved by using an imaging lens to collect the scattered light, and the greater the collection efficiency of the scattered light, the higher the clarity of the SPR image. We have experimentally demonstrated above that the clarity of the SPR image increases with decreasing the distance between the lens and the prism from $d = 7$ mm to $d = 4$ mm. The above experiments were performed using the HSPRM system with a NA = 0.4 lens. We also investigated the spectral SPR imaging performance of the HSPRM system with a NA = 0.2 lens. The spectral SPR images of a patterned ultrathin film of $TiO_2$ sputtered on the SPR chip were obtained by fixing the NA = 0.2 lens at a distance of $d = 2$ cm from the prism. As shown in Supplementary Fig. 7, the spectral SPR and greyscale SPR images obtained with NA = 0.2 lens are blurrier than those images obtained with the NA = 0.4 lens (see Figs. 2b and 3b). The comparison of SPR images obtained with NA = 0.2 and NA = 0.4 lenses demonstrated again that the spatial resolution of the HSPRM system is mainly determined by the NA of the imaging lens, and higher spatial resolution can be obtained by using a larger NA lens. As mentioned above, with an NA = 0.4 lens fixed at $d = 4$ mm from the prism, the lateral resolution of the HSPRM system was measured to be 2 μm and 1.2 μm in the directions parallel and perpendicular to the SPW vector, respectively, superior to most of existing SPRi sensors.

Our HSPRM system exhibits a unique mode of operation unlike any existing SPRi sensor, which enables the HSPRM system to generate large FOV, high lateral resolution SPR images and allows users to flexibly select the regions of interest in SPR images for hyperspectral microscopic analysis. With the unique mode of operation described above, our HSPRM system can be used as conveniently as a conventional microscope, except that the observed object under the objective is replaced by its SPR image. Compared with existing SPRi sensors, our HSPRM system have the following distinctive features: (1) it has a wide spectral range from 400 nm to 1000 nm, enabling to

use the gold-film SPR chip (RW ≥ 510 nm), the silver-film SPR chip (RW ≥ 400 nm), and the metal/dielectric PWR chip (RW ≥ 400 nm) for imaging and quantitative analysis; (2) it can provide three types of images, including spectral SPR image, greyscale SPR image at a desired wavelength, and RW image; (3) it has an optional FOV from 0.884 mm² to 0.003 mm², capable of observing the overall distribution of the sample and analyzing its local details, for example measuring single cells or cell populations without labeling; (4) our HSPRM system can provide single-pixel SPR radiance spectra with well-defined resonance valleys to improve the FOM of single-pixel spectral SPR sensor; (5) thanks to the hyperspectral imager used with up to 1456 spectral channels, our HSPRM system has high wavelength resolution ($\Delta\lambda = 0.41$ nm), enabling high-precision quantification based on single-pixel spectral SPR sensing; (6) it can be used to reconstruct the 2D thickness profiles of dielectric thin films using the measured RW image; (7) the time resolution of our HSPRM system depends on the number of pixels and that of spectral channels after the pixel fusion and spectral channel merging, and the minimum time resolution is 3.4 s, corresponding to the 176 spectral channels and 240 × 275 pixels. Namely, it requires 3.4 s to acquire a hyperspectral datacube containing 11.6 million data points. In our routine experiments with the HSPRM system, there is a trade-off between temporal resolution, spectral resolution and image resolution. The time required to acquire a hyperspectral datacube is typically 11 s, and each SPR spectrum contains 360 data points and each SPR image contains 480 × 550 pixels. Table 1 shows performance comparison of the HSPRM system with existing prism-based SPRi sensors. With the above-mentioned unique mode of operation, remarkable features and excellent performance, our HSPRM system has achieved a technological breakthrough in the field of SPR sensors, showing broad application prospects in label-free detection, imaging and quantification of chemical and biological measurands. Our HSPRM system is not limited to SPR-based label-free detection, but also has

**Table 1 | Performance comparison of advanced prism-based SPRi sensors**

|  | This work | Ref. 8 | Ref. 26 |
|---|---|---|---|
| Imaging mode | Hyperspectral imager with push-boom mode | CMOS and AOTF with wavelength scanning | CMOS camera with line scanning |
| Imaging speed | 80 fps | 20 fps |  |
| Spectral range | 400–1000 nm | 620–680 nm | 632 nm |
| Spectral channels | 1456 | 20 | 1 |
| Acquisition time | At least 3.4 s | At least 1 s |  |
| Spatial resolution |  |  |  |
| Parallel to SPW | 2 μm | 4.43 μm | 2.8 μm |
| Perpendicular to SPW | 1.2 μm | 3.96 μm | 1.7 μm |
| FOV |  |  |  |
| Parallel to SPW | 1250 μm | 297.6 μm |  |
| Perpendicular to SPW | 900 μm | 230.4 μm | ≤0.1 mm$^2$ |

potential applications for total internal reflection fluorescence (TIRF) hyperspectral microscopy.

## Methods

### Preparation of gold films and TiO$_2$ ultrathin films
In this work, the gold-film SPR chips used for RI detection and incident angle calibration were prepared by magnetron sputtering. First, the glass substrate was ultrasonically cleaned by acetone, ethanol, and deionized water in sequence, and then, a 3 nm chromium layer and a 44 nm gold film were consecutively sputtered on a 1-mm-thick glass substrate. The TiO$_2$ ultrathin film covered on the gold film was fabricated by continuous magnetron sputtering of TiO$_2$ on the gold film for 150 s.

### Fabrication of patterned SPR chip
The patterned SPR chip was fabricated with standard photolithography. First, the positive photoresist was spin-coated at speeds of 500 rpm for 5 s, 3000 rpm for 60 s and 500 rpm for 5 s on the gold film, forming a 1.5-μm-thick photoresist layer. Then, the chip was heated at 100 °C for 1 min and exposed under a patterned mask for 2 s. Finally, the chip was developed with 0.6% NaOH solution for 25 s to obtain the SPR chip with patterns formed in the exposed gold film regions.

### Immobilization of PS spheres on the gold film
The PS spheres with diameter of 0.6–1.0 μm (2.5% w/v) were purchased from Shanghai Aladdin, and dispersedly immobilized on the gold film by a reported method[38]. First, the gold-film chip was preheated at 160 °C for 10 min. Then, the PS spheres solution was diluted with alcohol to a suitable dispersion, and then dropped on the preheated gold film. After the solvent volatilized naturally, PS spheres were dispersed and immobilized on the gold film.

### Transfer of monolayer graphene to the gold film
Monolayer graphene deposited on the copper foil purchased from Graphenea SA (Spain), and was transferred on the gold film by a wet transfer method[39]. The preparation process includes the following steps: First, prepare the PMMA protective film by spin-coating a 2% PMMA acetone solution on the surface of the copper-based monolayer graphene sample. Then, put the pre-prepared PMMA/graphene/copper foil sandwich sample into a 3% ammonium persulfate solution to etch the copper foil, and rinse the sample with deionized water every 10 min. After the copper foil was dissolved, transfer the sample using a clean glass substrate to the deionized water to

remove the residual ammonium persulfate in the graphene. Next, slowly take out the sample from the deionized water with a 44 nm gold film-covered glass substrate, and place the sample at a certain angle for 30 min to completely evaporate the remaining water to ensure that the graphene and the substrate were closely attached. Finally, remove the PMMA protective film by immersing the sample in an acetone solution to obtain the monolayer graphene-covered SPR chip.

### Preparation of PMMA thin film
The microstructured PMMA film was prepared by spin coating and selectively dissolving PMMA/PS blends[40]. First, the PMMA/PS blend was prepared by mixing and dissolving PMMA and PS polymers in tetrahydrofuran. The solution concentration was 0.2% w/v, and the mass ratio of PMMA to PS component was 75%: 25%. Then, a 3-nm chromium layer and a 44-nm gold film were consecutively sputtered on a 1-mm-thick glass as the substrate, and 60 μL of PMMA/PS blend solution was spin-coated at speed of 500 rpm for 5 s, 3000 rpm for 60 s and 500 rpm for 5 s on the substrate to form a copolymer film. Next, the copolymer film was heat-cured at 100 °C for 2 h, and then naturally cooled and immersed in cyclohexane for 1 h. Finally, a microstructured PMMA film was obtained after cleaning and drying the copolymer film taken out from cyclohexane.

### Preparation of monolayer onion cell walls
The onion outer epidermal cells were torn from a yellow onion bulb purchased in the local store. First, peel off the outermost dry skin of the onion bulb and choose fresh onion scales close to the stem core. Then, drop the deionized water on the glass slide, after that, slowly tear off the single-cell layer of the outer epidermis of the onion with tweezers, and place it on the glass slide. Next, obtain the monolayer cell walls adhered to the gold film substrate by artificially disrupting the cell structure and multiple washing to lose cell contents. After that, put a cover glass and absorb the excess water with dust-free paper, leaving the sample stand for a while. After the water evaporated, the sample can be used for experimental observation. The sample was first observed using a dark-field microscope to select a region of interest, and then measured by our HSPRM system.

### Angular calibration and thickness determination
In this work, we carried out the wavelength-interrogated SPR sensing experiments using our HSPRM system. A beam of collimated and p-polarized white light is incident on the refracting face of the prism at a fixed angle θ to excite SPR. The reflected beam refracted out of the prism was received by the hyperspectral microscope through the specular reflection of the mirror (Fig. 1a). The single-pixel visible-NIR SPR spectrum of each pixel with wavelength separation of 0.41 nm was extracted from the acquired hyperspectral datacube. For calculation of the incident angle at each pixel, we use the three-layer (prism/gold-film/surrounding-medium) Fresnel formula expressed as:

$$R_{123} = |r_{123}|^2 = \left| \frac{r_{12} + r_{23}\exp(i2k_2 d_2)}{1 + r_{12}r_{23}\exp(i2k_2 d_2)} \right|^2 \qquad (1)$$

For calculation of the nanometric thin film thickness at each pixel, we use the four-layer Fresnel formula of a prism/gold film/measured film/medium expressed as:

$$R_{1234} = |r_{1234}|^2 = \left| \frac{r_{12} + r_{234}\exp(i2k_2 d_2)}{1 + r_{12}r_{234}\exp(i2k_2 d_2)} \right|^2 \qquad (2)$$

For calculation of the protein adlayer thickness at each pixel, we use the five-layer Fresnel formula of a prism/gold film/ dielectric film/

protein adlayer/medium expressed as:

$$R_{12345} = |r_{12345}|^2 = \left| \frac{r_{12} + r_{2345}\exp(i2k_2 d_2)}{1 + r_{12}r_{2345}\exp(i2k_2 d_2)} \right|^2 \qquad (3)$$

With

$$r_{2345} = \left| \frac{r_{23} + r_{345}\exp(i2k_3 d_3)}{1 + r_{23}r_{345}\exp(i2k_3 d_3)} \right|^2 \qquad (4)$$

$$r_{345} = \left| \frac{r_{34} + r_{45}\exp(i2k_4 d_4)}{1 + r_{34}r_{45}\exp(i2k_4 d_4)} \right|^2 \qquad (5)$$

$$r_{234} = \left| \frac{r_{23} + r_{34}\exp(i2k_3 d_3)}{1 + r_{23}r_{34}\exp(i2k_3 d_3)} \right|^2 \qquad (6)$$

$$r_{ij} = \frac{n_j^2 k_i^2 - n_i^2 k_j^2}{n_j^2 k_i^2 + n_i^2 k_j^2}, \qquad i = 1, 2, 3, 4; \quad j = 2, 3, 4, 5 \qquad (7)$$

$$k_i = \frac{2\pi}{\lambda}\left(n_i^2 - n_1^2 \sin^2\alpha\right)^{1/2}, \qquad i = 1, 2, 3, 4, 5 \qquad (8)$$

Where $\alpha = \pi/4 + a\sin(\sin(\theta)/n_1)$ is the total internal reflection angle at the metal/glass interface of the SPR chip; $\theta$ is the incident angle between the collimated beam and the normal of the prism's refracting face; $n_i$ is RI of the $i$th layer of the SPR chip, and the first layer is the glass substrate (the same material as the prism); $k_i$ is the wavevector component in the direction normal to the interface; $r_{ij}$ is the Fresnel reflection coefficient at the interface between the $i$th layer and the $j$th layer; $d_2$ is the thickness of the gold film; $d_3$ is the thickness of the dielectric film to be measured, including TiO$_2$ film, PMMA film, monolayer graphene, and cell wall. $d_4$ is the BSA molecules adlayer thickness to be measured.

In the calculation process, the above-mentioned Fresnel formula is used to simulate SPR/PWR spectrum and then using the simulated RW to fit the measured RW to determine the unknown quantity, including the incident angle $\theta$, the film thickness $d_3$, the adlayer thickness $d_4$, and the effective RI of cell wall. The prism RI ($n_1$) is RI of K9 glass or ZF7LA glass. The complex RI of gold film ($n_2$) was obtained from the published literature[41]. In the three-layer Fresnel formula, $n_3$ is RI of air or an aqueous solution. In the sensitivity detection (Fig. 5), RIs of aqueous glycerol solutions were measured at 23 °C by an abbe refractometer. In the four-layer Fresnel formula for calculating film thickness, RI of the TiO$_2$ or PMMA film ($n_3$) was obtained from the literatures[42,43], and $n_4$ is RI of air or an aqueous solution. In the four-layer Fresnel formula for calculation of cell wall thickness $d_3$ and its effective RI ($n_3$), at least two experimental RWs are fitted to simultaneously determine $n_3$ and $d_3$. In the five-layer Fresnel formula, $n_3$ is RI of the PMMA film, $n_4$ is RI of the BSA adlayer ($n_4$ = 1.429), and $n_5$ is RI of the aqueous solution of 50 μM BSA, which was measured to be $n_5$ = 1.3325 at 23 °C.

## Data collection and processing

For all measurements using the HSPRM system, the hyperspectral SPR datacubes were collected using the special SpecView software for the hyperspectral imager (GaiaField Pro-V10E, Dualix Spectral Imaging). Python and MATLAB were used to extract all RW images for quantitative analysis. MATLAB was used to calculate thickness and RI values according to the Fresnel formula. All data were processed and plotted with MATLAB.

## Reporting summary

Further information on research design is available in the Nature Research Reporting Summary linked to this article.

## Data availability

The raw data files generated in this study have been deposited at https://doi.org/10.6084/m9.figshare.21272856.v1. The processed data covered in this paper are collected in a file named Source Data provided with this paper. Source data are provided with this paper.

## Code availability

The code used to analyze the raw data generated in this study are available from the corresponding author upon request.

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

## Acknowledgements

This work was supported by the National Natural Science Foundation of China (62121003, 61931018, 61871365), the National Key R&D Program of China (2021YFB3200100), and the Scientific Instrument Developing Project of the Chinese Academy of Sciences (YJKYYQ20180004).

## Author contributions

Z.M.Q. conceived the concept, designed the HSPRM system, supervised the experiment and data analysis, and revised the manuscript. Z.W.L. prepared the samples, carried out the experiments, analyzed the experimental results, wrote and revised the manuscript. J.N.W. prepared the biosample, performed the biosample measurement and analyzed the bio-related results, Z.W.L. and J.N.W. and C.C. and B.Y. and Z.M.Q constructed the HSPRM system. All authors discussed the data and gave approval to the final version of the manuscript.

## Competing interests

The authors declare no competing interests.
