## [Peer Review File · Nature Communications]

Flexible hyperspectral surface plasmon resonance microscopyREVIEWER COMMENTS

Reviewer #1 (Remarks to the Author):

The authors of this paper have developed a new SPRM technology that combines a hyperspectral microscope and a Krestchmann SPR setup to realize single-pixel wavelength-interrogated SPR in parallel sensing and regioselective spectral SPR imaging, in which the problem of the divergence of the incident beam is solved using the Fresnel model based fitting algorithm, and the secondary imaging of L1 is used to match a hyperspectral microscope. They also demonstrated the possible applications of this technology by the detection of two-dimensional thickness distribution of different nanometric layers, quantitative spatial map of the amount of adsorbed protein, and the imaging of onion epidermal cells on the SPR substrate. Please see below my concerns:

1. Although it is a smart strategy to use the secondary imaging of L1 to couple a hyperspectral microscope, I think, the lateral resolution of this system is determined by the numerical aperture of L1, not by objective lens, thus resulting in loss of the lateral. What is the lateral resolution of this SPRM system?
2. What is the imaging speed of the hyperspectral imager used in this paper? The principle of the hyperspectral imager should be introduced in order to show the difference from the published SPRM system combining AOTF and white light source and CMOS camera (Wavelength-scanning surface plasmon resonance microscopy: A novel tool for real time sensing of cell-substrate interactions, Biosensors and Bioelectronics 145 (2019) 111717)?
3. How many points per SPR spectra in your experiments are used for region SPRM imaging? And how long did you obtain such a SPRM image in your experiments?
4. In terms of the detection of the cell wall, how do we distinguish the SPR signal response to the wall thickness or cellular components, because the shift of resonance wavelength is a combined contribution of them?
5. The electric field of the SP wave penetrates to a depth of approximately 200 nm at 630 nm in the direction perpendicular to the sensing film to the detection medium, so the micrometer order thickness of cell wall should be verified by other public technologies.

Reviewer #2 (Remarks to the Author):

The manuscript from Liu et al. presents a prism SPR imaging system coupled with hyperspectral analysis to improve the SPR imaging performance. It is a meaningful trial, because it can combine the wide detection range of spectral interrogation and spatially resolving capability of SPR imaging. However, the claims in current version are lack of solid support, and some of them conflict with common imaging techniques. In addition, the major advantage of wavelength interrogation seems not fully demonstrated. Thus, I cannot recommend the present version for publication, but it may be reconsidered after solving following concerns. I have the following specific questions and comments:

- 1) What is the major advantage of hyperspectral SPR imaging compared to single wavelength SPR imaging? I think the multi-wavelength detection could distinct different analyte based on their spectroscopy characteristics (Review of Scientific Instruments, 2001, 72, 3055-3060), for example, analyte with different absorption spectrum could be differentiated. But this major advantage is not demonstrated in this study.
- 2) The authors claimed that they can improve spatial resolution, as discussed in detail at Lines 185~189 "In order to solve this problem so that the spatial resolution is significantly improved." However, the data presented is not supporting the claim.
 - i) as shown in Fig.1, the hyperspectral microscopy analyzes the images formed by the lens L1. Thus, the spatial resolution has been determined by L1, whose aperture has determined the upper spatial frequency of collected signal light. As a simpler example, you cannot use high magnification zoom to improve the spatial resolution without changing the objective to a higher numerical aperture. This is

evidenced in Fig. 4b, where 10x and 20x objective can only zoom in the image, but not clearer images.

ii) The authors do not provide a quantitative result about the spatial resolution, but the data shown in Fig. 4 shows that the spatial resolution is ~5 micrometers. This is not better than the classical prism coupled SPR microscopy, which can provide spatial resolution of ~3 micrometers (Anal. Chem. 2007, 79, 7, 2979–2983).

iii) It seems that the authors also did not consider the delocalized surface plasmon propagation effect on the SPR imaging resolution. If one wants to develop high spatial resolution SPR imaging approaches, they should consider the delocalization of surface plasmon propagation along the surface (Phys. Rev. Lett. 78, 4269–4272 (1997)), such as use sensing film with short propagation length (Anal. Chem. 2007, 79, 7, 2979–2983), scanning localized SPR (Biomed. Opt. Express 3, 354–359 (2012)), and SPR scattering (Nat Methods 17, 1010–1017 (2020)).

3) Another major issue in this manuscript is the lack of some essential quantitative data to define the claimed performance of the instrument.

i) What are the numerical values of field of view, time resolution and spatial resolution?

ii) In line 69 “The system can provide SPR radiance spectrum in addition to the SPR intensity spectrum of each pixel in the HSPRM image to eliminate the system interference and improve the accuracy of measured resonance wavelength (RW);” How many times the accuracy has been improved?

iii) In line 71 “Local incident angle at each pixel in the HSPRM image is calibrated by the best fit between the simulated and measured single-pixel SPR spectra, solving the problem of incident beam divergence”. How much has the divergence level been improved to?

4) Fig. 7 presents the onion cells on the gold surface. Do Fig. 7a and 7b come from the same area?

5) Line 69 claimed the technology is suitable for “in vivo imaging”. “In vivo” usually means that imaging in a living systems or animals, which is clearly beyond the capabilities of prism SPR systems.

Responses to Reviewers' comments:

First of all, we greatly appreciate your constructive comments and advices, which are very helpful for improving manuscript during our revision process. Our point-to-point responses to your comments are as follows.

From Reviewer #1:

The authors of this paper have developed a new SPRM technology that combines a hyperspectral microscope and a Kretschmann SPR setup to realize single-pixel wavelength-interrogated SPR in parallel sensing and regioselective spectral SPR imaging, in which the problem of the divergence of the incident beam is solved using the Fresnel model based fitting algorithm, and the secondary imaging of L1 is used to match a hyperspectral microscope. They also demonstrated the possible applications of this technology by the detection of two-dimensional thickness distribution of different nanometric layers, quantitative spatial map of the amount of adsorbed protein, and the imaging of onion epidermal cells on the SPR substrate. Please see below my concerns:

Comment 1: Although it is a smart strategy to use the secondary imaging of L1 to couple a hyperspectral microscope, I think, the lateral resolution of this system is determined by the numerical aperture of L1, not by objective lens, thus resulting in loss of the lateral. What is the lateral resolution of this SPRM system?

Response: Thanks for the comment and question. We agree with that the lateral resolution of the HSPRM system is determined by the numerical aperture (NA) of imaging lens L1. We have added new experimental results of spatial resolution and demonstrated that in the revised manuscript. The experimental comparison of SPR images obtained with NA = 0.2 and NA = 0.4 imaging lenses demonstrated that higher spatial resolution was obtained by using a higher NA lens. To demonstrate the spatial resolution of the HSPRM system with the imaging lens (NA = 0.4), we used polystyrene (PS) spheres of less than 1 μm in diameter as the point objects immobilized on the gold film for SPR imaging. Figure R1a shows the SPR greyscale image at $\lambda = 592$ nm, extracted from the measured hyperspectral SPR datacube. The dark background of the image indicates that SPR occurs on the surface of the gold film at $\lambda = 592$ nm. The image contains some bright spots surrounded by bright rings, representing the isolated PS spheres and their Airy disk pattern. Figure R1b and R1c show the x-axis and y-axis normalized intensity profiles of a smaller PS spot in the image and their best fitting curves with Gaussian function, and the y-axis is the propagation direction of surface plasmon wave (SPW). The full width at half maximum (FWHM) of the intensity peak in the best fitting curve represents the lateral resolution¹. The resulting lateral resolution is determined to be **1.2 μm** in the directions perpendicular to the SPW vector and **2 μm** in the directions parallel to the SPW vector, which are superior to most of existing prism-based SPRi sensors²⁻⁴. We have added this result

to the Results section (Lines 186~209) and Discussion section (Lines 371~379) in the revised manuscript.

Figure R1. **a** SPR greyscale image at $\lambda = 592$ nm extracted from the measured hyperspectral SPR datacube of the SPR chip with PS spheres less than $1 \mu\text{m}$ in diameter immobilized on the surface. **b** Lateral resolution of $1.2 \mu\text{m}$ perpendicular to the SPW propagation direction. **c** Lateral resolution of $2.0 \mu\text{m}$ parallel to the SPW propagation direction.

Comment 2: What is the imaging speed of the hyperspectral imager used in this paper? The principle of the hyperspectral imager should be introduced in order to show the difference from the published SPRM system combining AOTF and white light source and CMOS camera (Wavelength-scanning surface plasmon resonance microscopy: A novel tool for real time sensing of cell-substrate interactions, Biosensors and Bioelectronics 145 (2019) 111717)?

Response: Thanks for the question and suggestion. The imaging speed of the hyperspectral imager used in our work is up to 80 frames per second (fps), faster than the CMOS-AOTF combination (20 fps).

Figure R2 demonstrates the difference between our hyperspectral imager and the published SPRM system combining AOTF and white light source and CMOS camera. Figure R2a shows the imaging principle of the published SPRM system combining AOTF and white light source and CMOS camera. AOTF is used to filter the white light, and CMOS camera is used to collect spatial information of the target at each spectral channel. By wavelength scan of the AOTF in the spectral dimension, grayscale images of all spectral channels are captured. Then, the spectral information of each pixel is obtained by fitting the single-pixel intensity of the image in each spectral channel. This method achieves short acquisition time (1 s) but moderate spectral range (620-680 nm), and there are only 20 data points per spectrum³.

The imaging principle of the hyperspectral imager of our HSPRM system is shown in Figure R2b. There is an entrance slit and a prism-grating-prism dispersive element in the hyperspectral imager, and it first acquires spatial and spectral information of line-arranged (Y direction) 1936×1 pixels at a time and arrange them on a 2D detector. Then a spectral image with 1936×2202 pixels and spectra at all pixels is obtained by line scanning in the spatial dimension (X direction). The spectral image is stored as a hyperspectral datacube. This method has a wider spectral range (400-1000 nm) and more data points per spectrum (1456) than the published SPRM system combining AOTF and white light source and CMOS camera. The temporal resolution of our hyperspectral imager is limited by the push-broom mode, which can be improved by internal pixel fusion and spectral channel merging, and externally by using a high-power light source. In this work, we used a high-brightness LDLS EQ-99 lamp as the light source, which enables the hyperspectral imager to acquire a single-pixel spectrum in only 0.5 ms and to acquire images at an imaging speed of 80 fps. In this case, the prepared HSPRM system requires a minimum time of 3.4 s to acquire a hyperspectral SPR datacube with pixel fusion and spectral channel merging. This optimal temporal resolution is a result of sacrificing the image resolution and spectral resolution. In our experiments, a trade-off is often made between temporal resolution, spectral resolution and image resolution. We have included this information in the Introduction section (Lines 49~51) and Results section (Lines 96~106) in the revised manuscript.

Figure R2. **a** Imaging principle of the published SPRM system combining AOTF and white light source and CMOS camera. **b** Imaging

principle of the hyperspectral imager of our HSPRM system.

Comment 3: How many points per SPR spectra in your experiments are used for region SPRM imaging? And how long did you obtain such a SPRM image in your experiments?

Response: Thanks for the question. Our HSPRM system can provide visible-near infrared single-pixel SPR spectra with 1456 points per spectrum and SPR images with up to 1936×2202 pixels. The time resolution of the HSPRM system depends on the number of pixels and the points per SPR spectrum. With the pixel fusion and spectral merging, the minimum time resolution is 3.4 s, corresponding to the 176 points per single-pixel SPR spectrum and 240×275 pixels in the SPR image. Namely, it requires 3.4 s to acquire a hyperspectral datacube containing 11.6 million data points. In our routine experiments with the HSPRM system, there is a trade-off between temporal resolution, spectral resolution and image resolution. The time required to acquire a hyperspectral datacube is typically 11 s, and each SPR spectrum contains 360 data points and each SPR image contains 480×550 pixels. We have included this information in the Introduction section (Lines 62~64) and Discussion section (Lines 393~398) in the revised manuscript.

Comment 4: In terms of the detection of the cell wall, how do we distinguish the SPR signal response to the wall thickness or cellular components, because the shift of resonance wavelength is a combined contribution of them?

Response: Thanks for the question and discussion. To demonstrate whether the SPR signal response is to wall thickness or cellular composition, we added new experimental results of cell wall detection. Specifically, we peeled a monolayer of onion epidermal cell walls from an onion and attached it to the gold film of the SPR chip for use as a biosample by artificially disrupting the cell structure and washing multiple times to wash away cell contents. Figure R3a shows a dark-field optical microscope image of the biosample, and the onion epidermal cell walls are clearly seen in the image. Figure R3b is the spectral SPR image of the same cell area with contours consistent with the dark-field image. The yellowish and green regions in the spectral SPR image, corresponding to the dark and light regions in the dark-field microscope image, represent the cell wall attached area (without air gap) and cell wall non-attached area (with air gap).

Onion cell wall is mainly composed of cross-linked polysaccharides such as cellulose, hemicellulose, and pectin (refractive index (RI) is about 1.5), and the cell wall thickness is in the micrometer order, thus it can act as a waveguide layer covered with metal surface to excite plasmon waveguide resonance (PWR)⁵⁻⁷. Figure R3c shows the spectrum of pixel A selected in the cell wall non-attached area, which contains a single resonance valley, corresponding to the SPR mode, and the resonance wavelength (RW) is close to that of the gold film region. Figure R3d shows the spectrum of pixel B selected in the cell wall attached area, which contains four well-defined resonance valleys, indicating that the cell wall forms the PWR structure together with the gold film. The PWR spectrum of pixel B was fitted by using the spectrum simulation based on the four-layer Fresnel model, resulting in the cell wall thickness of 1.973 μm and its effective RI of

1.542 at pixel B. The obtained cell wall thickness is close to the reported value of $1.62 \pm 0.36 \mu\text{m}^8$. Furthermore, we derived the 2D RW distribution from the measured hyperspectral SPR datacube of the biosample, and then obtain the 2D thickness and effective RI distributions of the cell walls, as shown in Figure R3e and Figure R3f. The above experimental results show that our HSPRM system is very efficient and informative for cell analysis, and it can clearly distinguish the degree of adhesion of cells to the substrate, and its two-dimensional quantification capability makes it superior to existing SPRi sensors. We have added this result to the Results section (Lines 328~353) in the revised manuscript.

Figure R3. **a** Surface morphology of monolayer onion cell walls adhered to a gold film substrate observed by the dark-field microscopy. **b** Spectral SPR image of the same cell area. **c** Single-pixel SPR spectra of the gold film substrate and the cell wall area that is not attached to the substrate. **d** Experimental and simulated PWR spectra of the cell wall area that is attached to the substrate. **e** 2D thickness distribution of the cell walls that are attached to the substrate. **f** 2D effective RI distribution of the cell walls that are attached to the substrate. Scale bar = 50 μm .

Comment 5: The electric field of the SP wave penetrates to a depth of approximately 200 nm at 630 nm in the direction perpendicular to the sensing film to the detection medium, so the micrometer order thickness of cell wall should be verified by other public technologies.

Response: Thanks for the suggestion. We agree that the micrometer order thickness of cell wall goes beyond the penetrated depth of SP wave, and should be verified by other public technologies. We have prepared another biosample of monolayer onion cell wall on the gold film substrate, and obtain the 2D thickness and effective RI distribution of the cell walls that are attached to the substrate by PWR spectroscopy. The experimental and calculated results are demonstrated in the above Figure R3, and we also updated this result in the revised manuscript (Lines 328~353).

From Reviewer #2:

The manuscript from Liu et al. presents a prism SPR imaging system coupled with hyperspectral analysis to improve the SPR imaging performance. It is a meaningful trial, because it can combine the wide detection range of spectral interrogation and spatially resolving capability of SPR imaging. However, the claims in current version are lack of solid support, and some of them conflict with common imaging techniques. In addition, the major advantage of wavelength interrogation seems not fully demonstrated. Thus, I cannot recommend the present version for publication, but it may be reconsidered after solving following concerns. I have the following specific questions and comments:

Comment 1: What is the major advantage of hyperspectral SPR imaging compared to single wavelength SPR imaging? I think the multi-wavelength detection could distinct different analyte based on their spectroscopy characteristics (Review of Scientific Instruments, 2001, 72, 3055-3060), for example, analyte with different absorption spectrum could be differentiated. But this major advantage is not demonstrated in this study.

Response: Thanks for the question and suggestion. Single-wavelength SPR imaging with the intensity interrogation has a limited detection dynamic range because the image intensity varies within a limited range with the sample refractive index (RI) and/or thickness and can easily reach saturation. Compared to the single-wavelength SPR imaging, hyperspectral SPR imaging with the wavelength interrogation is more informative and has a wider detection dynamic range. We have included this information in the Introduction section (Lines 35~53) in the revised manuscript. Specifically, our HSPRM system can perform spatial imaging under up to 1456 individual spectral channels within the spectral range from 400 nm to 1000 nm and provide greyscale SPR images of each spectral channel, like working as single-wavelength SPR imaging. On the other hand, our HSPRM system can obtain the spectral SPR image of the measured sample and perform spectral analysis of all individual pixels in the image, like miniature spectrometers working in parallel. It can provide visible-near infrared SPR spectrum of each pixel, allowing to distinct different analyte based on the spectroscopy characteristics⁹. We may experimentally demonstrate this advantage in future work, but this is not the focus of this work, and we greatly appreciate and acknowledge the reviewer for suggesting this advantage.

Besides, unlike any existing SPRi sensor, our HSPRM system operates in a unique mode involving two steps: first generating a spectral SPR image of the sample using the achromatic imaging lens L1, and then analyzing the region of interest of this image using the hyperspectral microscope. Compared with existing SPRi sensors, our HSPRM system has the following distinctive features: (1) it can provide three types of images, including spectral SPR image, greyscale SPR image at a desired wavelength, and resonance wavelength (RW) image; (2) it has a wide spectral range from 400 nm to 1000 nm, enabling to use the gold-film SPR chip ($RW \geq 510$ nm), the silver-film SPR chip ($RW \geq 400$ nm), and the metal/dielectric plasmon waveguide resonance (PWR) chip ($RW \geq 400$ nm) for imaging analysis; (3) it has the lateral resolution of 2 μm and 1.2 μm in the directions parallel and perpendicular to the surface plasmon wave (SPW) vector,

respectively, superior to most of existing SPRi sensors; (4) it has an optional field of view (FOV) from 0.884 mm² to 0.003 mm², capable of observing the overall distribution of the sample and analyzing its local details, for example measuring single cells or cell populations without labeling; (5) it can provide single-pixel SPR radiance spectra with well-defined resonance valleys to improve the figure of merit of single-pixel spectral SPR sensor; (6) thanks to the hyperspectral imager used with up to 1456 spectral channels, our HSPRM system has high wavelength resolution ($\Delta\lambda = 0.41$ nm), enabling high-precision quantification based on single-pixel spectral SPR sensing; (7) it can be used to reconstruct the two-dimensional (2D) thickness profiles of dielectric thin films using the measured RW image.

RW image is the distribution of RWs extracted from the SPR spectrum of each pixel in the spectral SPR image. The RW image relies on RI and/or thickness of the sample and it can be used for quantification of samples based on Fresnel theory. Figure R1 displays the greyscale SPR image and RW image extracted from the same hyperspectral SPR datacube of a patterned SPR chip measured with our HSPRM system. The darkfield background regions in Figure R1a is positive photoresist-covered area, and the three regions marked A, B and C are the exposed gold film areas with the equal actual sizes of 5 $\mu\text{m} \times 35 \mu\text{m}$. Figure R1a shows two normalized intensity profiles along the x-axis and y-axis for the three marked regions. Figure R1b shows two corresponding RW intensity profiles of the three regions. These RW profiles exhibit abrupt changes without gradients at the edges of the marked regions, much better than the corresponding normalized intensity profiles in Figure R1a. The comparison in Figure R1 shows that the RW image is superior to the greyscale SPR image in terms of image quality and dimensional accuracy. We have experimentally demonstrated the advantages of our HSPRM system in Results section (Lines 116~353) and described in Discussion section (Lines 355~402) in the revised manuscript.

Figure R1. **a** Greyscale SPR image and two normalized intensity profiles. **b** RW image and two RW profiles. Both the greyscale image

and RW image are extracted from the same hyperspectral SPR datacube measured with our HSPRM system. All the normalized intensity profiles and RW profiles are for the three regions marked A, B, and C with actual sizes of $5\ \mu\text{m} \times 35\ \mu\text{m}$.

Comment 2: The authors claimed that they can improve spatial resolution, as discussed in detail at Lines 185~189 “In order to solve this problem so that the spatial resolution is significantly improved.” However, the data presented is not supporting the claim.

Response: Thanks for the question. We have modified the statements and added new experimental results of spatial resolution in the revised manuscript. In our HSPRM system, the p-polarized collimated beam is refracted into the prism at normal incidence and undergoes total internal reflection at the metal/glass interface to excite SPW on the metal surface. The reflected and scattered light is refracted out of the prism and then collected by an imaging lens L1 (NA = 0.4), resulting in a clear SPR image. Without the imaging lens L1, the triangle prism can output the SPR image contained in the reflected collimated beam, but the edges of the SPR image are blurred. The clarity of the SPR image can be improved by using the imaging lens to collect the scattered light, and the greater the collection efficiency of the scattered light, the higher the clarity of the SPR image, as shown in Figure R2 and R3. We have added this result to the Results section (Lines 116~185) in the revised manuscript.

We fabricated a patterned SPR chip by standard photolithography using a $1.5\text{-}\mu\text{m}$ -thick positive photoresist layer for SPR imaging with the HSPRM system. Figures R2a and R2b each show three spectral SPR images obtained with 5x, 10x, 20x objectives. The distance from the lens L1 to the prism’s refracting surface is $d = 7\ \text{mm}$ in Figures R2a and $d = 4\ \text{mm}$ in Figures R2b. In each image, the exposed gold film area (green) and the photoresist-covered region (red) can be clearly distinguished with two different colors. The combination of Figures R2a and R2b indicates that our HSPRM system can easily provide spectral SPR images with different FOVs by using objectives of different magnifications. A comparison of Figures R2a and R2b reveals that the FOV of the HSPRM system can also be effectively adjusted by changing the distance of the lens L1 from the prism’s refracting surface. It is worth noting that, in Figures R2b, the quality of the spectral SPR image of two $1.6\ \mu\text{m}$ -wide parallel bars obtained with the 20x objective is degraded by the insufficient light intensity for the given exposure time.

Figures R3a and R3b show two greyscale SPR images at the wavelength of 515 nm, extracted from the 20x spectral image in Figures R2a and the 5x spectral image in Figures R2b, respectively. We select these two images for comparison because their FOVs are close to each other. The greyscale image in Figures R3b is clearer than that in Figures R3a, as seen with naked eye. The other two plots in Figures R3a show the normalized intensity distributions along the x-axis and y-axis for the three regions marked A, B, and C in the greyscale image, respectively. The three regions marked correspond to the exposed gold film areas with the equal actual sizes of $5\ \mu\text{m} \times 35\ \mu\text{m}$. As can be seen from the two normalized intensity profiles, the dimensions of the three marked regions determined by the greyscale image are all larger than the

actual size in the x-axis and y-axis directions. The blurred edges of each marked region reveal the insufficient lateral resolution of the lens L1. Figure R3b also includes the two normalized intensity distribution plots for the same regions as those in Figure R3a. The size of each region obtained in Figure R3b is larger than its actual size but smaller than the corresponding value obtained in Figure R3a. This comparison combined with the result in Figure R2 verified that increasing the distance of the lens L1 from the prism's refracting surface can result in an increased FOV but a reduced lateral resolution. Therefore, in this work, the lens L1 is typically mounted $d = 4$ mm from the refracting surface of the prism for routine high-resolution spectral SPR imaging measurements.

Figure R2. **a** Three spectral SPR images of a patterned SPR chip measured using the HSPRM system with a NA = 0.4 imaging lens fixed at $d = 7$ mm from the prism. The FOVs of three images corresponding to the 5x, 10x and 20x objectives are 0.884 mm^2 , 0.145 mm^2 and 0.042 mm^2 , respectively. **b** Similar results measured in the case of fixing the NA = 0.4 imaging lens at $d = 4$ mm from the prism. The FOVs of three SPR images are 0.049 mm^2 , 0.010 mm^2 , and 0.003 mm^2 , respectively.

Figure R3. **a** Grayscale image at 515 nm wavelength and normalized intensity distributions along the x- and y-axes for the regions labeled A, B and C in the image. The grayscale image was extracted from the hyperspectral datacube acquired with the 20x objective and the NA = 0.4 imaging lens fixed at $d = 7$ mm from the prism. **b** Similar results obtained with the 5x objective and fixing the NA = 0.4 imaging lens at $d = 4$ mm from the prism.

i) as shown in Fig.1, the hyperspectral microscopy analyzes the images formed by the lens L1. Thus, the spatial resolution has been determined by L1, whose aperture has determined the upper spatial frequency of collected signal light. As a simpler example, you cannot use high magnification zoom to improve the spatial resolution without changing the objective to a higher numerical aperture. This is evidenced in Fig. 4b, where 10x and 20x objective can only zoom in the image, but not clearer images.

Response: Thanks for the suggestion and bringing the spatial resolution issue to our attention. We agree with that the lateral resolution is determined by L1 not the objective. The experimental comparison of SPR images obtained with NA = 0.2 and NA = 0.4 imaging lenses demonstrated that higher spatial resolution was obtained by using a higher NA lens. We also find that the imaging lens L1 should be fixed close to the prism to effectively collect scattered light, thereby improving the SPR grayscale image quality, as shown in Figure R3. In our HSPRM system, the lens L1 (NA = 0.4) has the focal length of 15 mm and is typically mounted ~4 mm from the prism for routine high-resolution spectral SPR imaging measurements. The size of our isosceles right-angle prism with a hypotenuse length of 36.8 mm limits the use of a shorter focal length L1 to obtain higher NA. To demonstrate the spatial resolution of the HSPRM system with the imaging lens L1 (NA = 0.4), we used polystyrene (PS) spheres of less than 1 μm in diameter as the point objects immobilized on the gold film for SPR imaging. Figure R4a shows the SPR grayscale image at $\lambda = 592$ nm, extracted from the measured hyperspectral SPR datacube. The dark background of the image indicates that SPR occurs on the surface of

the gold film at $\lambda = 592$ nm. The image contains some bright spots surrounded by bright rings, representing the isolated PS spheres and their Airy disk pattern. Figure R4b and R4c show the x-axis and y-axis normalized intensity profiles of a smaller PS spot in the image and their best fitting curves with Gaussian function, and the y-axis is the SPW propagation direction. The full width at half maximum (FWHM) of the intensity peak in the best fitting curve represents the lateral resolution¹. The ultimately lateral resolution of our HSPRM system is determined to be 1.2 μm perpendicular and 2 μm parallel to the SPW propagation direction. We have included this information in Results section (Lines 186–209) and Discussion section (Lines 368–379) in the revised manuscript.

Figure R4. **a** SPR greyscale image at $\lambda = 592$ nm extracted from the measured hyperspectral SPR datacube of the SPR chip with PS spheres less than 1 μm in diameter immobilized on the surface. **b** Lateral resolution of 1.2 μm perpendicular to the SPW propagation direction. **c** Lateral resolution of 2.0 μm parallel to the SPW propagation direction.

ii) The authors do not provide a quantitative result about the spatial resolution, but the data shown in Fig. 4 shows that the spatial resolution is ~ 5 micrometers. This is not better than the classical prism coupled SPR microscopy, which can provide spatial resolution of ~ 3 micrometers (Anal. Chem. 2007, 79, 7, 2979–2983).

Response: Thanks for the question. As mentioned above, with imaging lens L1 (NA = 0.4), the lateral resolution of our HSPRM system was measured to be 2 μm and 1.2 μm in the directions parallel and perpendicular to the SPW vector, respectively, superior to most of existing SPRi sensors²⁻⁴.

iii) It seems that the authors also did not consider the delocalized surface plasmon propagation effect on the SPR imaging resolution. If one wants to develop high spatial resolution SPR imaging approaches, they should consider the delocalization of surface plasmon propagation along the surface (Phys. Rev. Lett. 78, 4269–4272 (1997)), such as use sensing film with short propagation length (Anal. Chem. 2007, 79, 7, 2979–2983), scanning localized SPR (Biomed. Opt. Express 3, 354-359 (2012)), and SPR scattering (Nat Methods 17, 1010–1017 (2020)).

Response: Thanks for the suggestion. Our HSPRM system utilizes the imaging lens to collect both scattered and reflected light from the SPR chip to form a SPR image with a large FOV, and then use the hyperspectral microscope to magnify the selected region of interest in the SPR image for spatially resolved quantitative analysis. The scattered light results from the out-of-plane scattering of the SPW at the edges of the object under study¹⁰, which means that at the scattering points at the object's edges, SPW is directly converted into photons radiated from the SPW propagation plane. The collected photons are beneficial for improving the clarity of SPR image of the object because the scattering points at the object's edges can be traced back along their propagation paths, and the greater the collection efficiency of the scattered light, the higher the clarity of the SPR image, as shown in Figure R2 and R3. As shown in Figure R3, due to the contribution of SPW propagation length, the difference between the measured and actual size of the region is larger in the y-axis direction ($\Delta L_y \geq 6 \mu\text{m}$) than in the x-axis direction ($\Delta L_x \leq 3 \mu\text{m}$). In addition, ΔL_y in Figure R3b is shorter than that in Figure R3a, suggesting that the influence of the SPW propagation length on the SPR image can be weakened by increasing the lateral resolution of the HSPRM system.

In fact, the lateral resolution is mainly limited by NA of imaging lens L1, surface plasmon propagation length, and geometric aberrations of the prism². Using the imaging lens L1 (NA=0.4) to collect more signal light and the isosceles right-angle prism with normal incidence to reduce geometric aberrations, we achieved the lateral resolution of 1.2 μm perpendicular and 2 μm parallel to the surface plasmon propagation directions on a smooth metallic SPR surface. The surface plasmon propagation length should be shorter for better lateral resolution, but both the propagation length and sensitivity increase with wavelength. Moreover, decreasing the propagation length means increasing the imaginary part of the plasmon wave vector, which eventually results in a wider SPR curve and lower image contrast¹¹. Therefore, there is a trade off between them.

Other high spatial resolution SPR imaging strategies, such as designing nanostructured SPR surfaces or using other metal sensing films to reduce propagation length^{4,12}, scanning localized SPR¹³, top-view SPR scattering imaging of SPR chip surface¹⁴, will be considered in follow-up work but is not the focal point of this work. We would like to emphasize that the combination of hyperspectral microscope with spectral SPR imaging leads to a technological breakthrough in the field of SPR sensors and shows broad application prospects in label-free detection, imaging and quantification of chemical and biological measurands. We hope that vast possibilities and applications of the versatile HSPRM system could be

explored in the future.

Comment 3: Another major issue in this manuscript is the lack of some essential quantitative data to define the claimed performance of the instrument.

Response: Thanks for the question. We have added the essential quantitative data to the Discussion section (Lines 377~399) in the revised manuscript (see Table 1 as below).

Table 1 System performance comparison of advanced prism-based SPRM systems.

		This work	Reference^{3, 15}	Reference²
Imaging mode		Hyperspectral imager with push-boom mode	CMOS and AOTF with wavelength scanning	CMOS camera with line scanning
Imaging speed		80 fps	20 fps	
Spectral range		400~1000 nm	620~680 nm	632 nm
Spectral channels		1456	20	1
Acquisition time		At least 3.4 s	At least 1 s	
Spatial resolution	Parallel to SPW	2 μm	4.43 μm	2.8 μm
	Perpendicular to SPW	1.2 μm	3.96 μm	1.7 μm
Field of view	Parallel to SPW	1250 μm	297.6 μm	$\leq 0.1 \text{ mm}^2$
	Perpendicular to SPW	900 μm	230.4 μm	

i) What are the numerical values of field of view, time resolution and spatial resolution?

Response: As shown in Figure R2, the FOV of our HSPRM system ranges from 0.884 mm² to 0.003 mm², which is achieved by adjusting the distance of L1 from the prism's refracting surface, or using objectives of different magnifications. This optional FOV allows to measure single cells or cell populations without labeling.

The time resolution of our HSPRM system is limited by the push-broom mode of the hyperspectral imager, but it can be improved by internal pixel fusion and spectral channel merging, and externally by using a high-power light source. In this work, we used a high-brightness LDLS EQ-99 lamp as the light source, which enables the hyperspectral imager to acquire a single-pixel spectrum in only 0.5 ms and to acquire images at an imaging speed of 80 fps. In this case, the prepared HSPRM system requires a minimum time of 3.4 s to acquire a hyperspectral SPR datacube with pixel fusion and spectral channel merging. This optimal temporal resolution is a result of sacrificing the image resolution and spectral resolution. In our experiments, a trade-off is often made between temporal resolution, spectral resolution and image resolution. Generally, the time required to acquire a hyperspectral datacube is typically 11 s, and each SPR spectrum contains 360 data points and each SPR image contains 480 \times 550 pixels.

The lateral resolution of the HSPRM system was measured to be 2 μm and 1.2 μm in the directions parallel and perpendicular to the SPW vector, respectively, superior to most of existing SPRi sensors²⁻⁴. The system performance

comparison of advanced prism-based SPRM systems is shown in Table 1.

ii) In line 69 “The system can provide SPR radiance spectrum in addition to the SPR intensity spectrum of each pixel in the HSPRM image to eliminate the system interference and improve the accuracy of measured resonance wavelength (RW);” How many times the accuracy has been improved?

Response: Radiometric correction is an essential image-processing function of a hyperspectral imager for remote sensing^{16, 17}, which involves subtracting the background signal (bias) and dividing by the gain of the instrument and consequently converting the raw instrument output (digital numbers) to radiance. Figure of merit (FOM) is an important quantity used to evaluate the performance of SPR sensors, which is defined as a ratio of the sensor’s RI sensitivity to the FWHM of the SPR resonance valley¹⁸. Figure R5a and R5b show the RI-dependent SPR intensity spectra and the corresponding radiance spectra. The resonance valleys in the SPR radiance spectra are much better in shape than those in the original SPR intensity spectra, meaning that the resonance wavelength can be more accurately determined from the SPR radiance spectrum than from the SPR intensity spectrum. With the SPR radiance spectrum, the FOM is obtained to be 44.8 RIU^{-1} , larger than that obtained with the SPR intensity spectrum ($\text{FOM} = 31 \text{ RIU}^{-1}$), as shown in Figure R5c. The SPR radiance spectrum can effectively enhance the FOM of the single-pixel spectral SPR sensor. We have included this analysis in Results section (Lines 228~232) in the revised manuscript.

Figure R5. a Experimental single-pixel SPR intensity spectra measured at a series of glycerol aqueous solutions with RI ranging from 1.3266 to 1.3491. **b** Single-pixel SPR radiance spectra obtained by radiometric correction. **c** Comparison of FOMs determined from SPR intensity spectrum and SPR radiance spectrum.

iii) In line 71 “Local incident angle at each pixel in the HSPRM image is calibrated by the best fit between the simulated and measured single-pixel SPR spectra, solving the problem of incident beam divergence”. How much has the divergence level been improved to?

Response: The calibrated angle of incidence at each pixel of the SPR image was obtained by fitting the measured single-pixel RI sensitivity using the three-layer Fresnel formula. As shown in Figure R6a, the calibrated incident angle increases quasi-linearly from 39.80° to 40.07° as the number of pixels along the y-axis increases, which means that the incident

collimated beam impinging on the $\sim 4 \text{ mm}^2$ image FOV has a divergence angle of 0.27° .

Using the pixel-by-pixel calibrated incident angles, the HSPRM system enables accurate 2D quantification of nanometric measurands. To demonstrate this effect, a large-area uniform ultrathin film of titanium dioxide (TiO_2) was sputtered on the gold film SPR chip for imaging analysis using the HSPRM system. Figure R6b and R6c shows the 2D thickness distribution in a $\sim 2 \text{ mm}^2$ FOV without and with angular calibration, respectively. It can be seen from Figure R6c that the film thickness is flatly distributed with an average thickness of 3.5 nm and a fluctuation of $\pm 0.1 \text{ nm}$, but in Figure R6b, the film thickness exhibits a tapered distribution along the y-axis with a minimum thickness of $3.3 \text{ nm} \pm 0.1 \text{ nm}$ and a maximum thickness of $3.6 \text{ nm} \pm 0.1 \text{ nm}$. Such tapered thickness distribution is related to the quasilinear divergence of the incident collimated beam. The standard deviation of the film thickness is 0.1 nm before the angular calibration and 0.05 nm after the angular calibration. The measurement precision increases 2 times with the angular calibration.

It is worth noting that the standard deviation of film thickness due to the incident beam divergence is highly dependent on the film RI, and the lower the film RI, the larger the standard deviation. We simulated the thickness distribution along the y-axis of an adsorbed bovine serum albumin (BSA) layer (thickness = 3.5 nm and RI = 1.429) using the above calibrated incident angles (Figure R6a). The simulation results show a tapered distribution of BSA film thickness along the y-axis with a minimum thickness of 2.8 nm and a maximum thickness of 4.3 nm in a $\sim 2 \text{ mm}^2$ FOV. The simulated standard deviation of the BSA adlayer is 0.36 nm, which is 3.6 times as large as that of the 3.5-nm-thick TiO_2 film measured above, evidencing that the standard deviation of film thickness increases with decreasing the film RI. Therefore, it is concluded that even a small divergence of the incident angle can seriously affect the quantified results of biochemical samples and thus the angular calibration is very important. We have included this analysis in Results section (Lines 248~271) in the revised manuscript.

Figure R6. **a** Actual incident angle at each pixel in the SPR image along in x and y-axis directions. **b** 2D thickness distribution of the TiO_2 thin film calculated without angular calibration. **c** 2D thickness distribution of the TiO_2 thin film calculated with angular calibration.

Comment 4: Fig. 7 presents the onion cells on the gold surface. Do Fig. 7a and 7b come from the same area?

Response: Yes.

Comment 5: Line 69 claimed the technology is suitable for “in vivo imaging”. “In vivo” usually means that imaging in a living systems or animals, which is clearly beyond the capabilities of prism SPR systems.

Response: Thanks for the suggestion. We followed the reviewer’s suggestion and removed the “in vivo imaging” in the revised manuscript.

References

1. Xu, L.-J., Lin X., He Q., Worku M. & Ma B. Highly efficient eco-friendly X-ray scintillators based on an organic manganese halide. *Nat. Commun.* **11**, 4329 (2020).
2. Laplatine, L. et al. Spatial resolution in prism-based surface plasmon resonance microscopy. *Opt. Express* **22**, 22771-22785 (2014).
3. Zeng, Y. J., Zhou J., Wang X. L., Cai Z. W. & Shao Y. H. Wavelength-scanning surface plasmon resonance microscopy: A novel tool for real time sensing of cell-substrate interactions. *Biosens. Bioelectron.* **145**, 111717 (2019).
4. Huang, B., Yu F. & Zare R. N. Surface plasmon resonance imaging using a high numerical aperture microscope objective. *Anal. Chem.* **79**, 2979-2983 (2007).
5. Qi, Z.-M., Honma I. & Zhou H. Nanoporous leaky waveguide based chemical and biological sensors with broadband spectroscopy. *Appl. Phys. Lett.* **90**, 011102 (2007).
6. Qi, Z. M., Wei M. D., Matsuda H., Honma I. & Zhou H. S. Broadband surface plasmon resonance spectroscopy for determination of refractive-index dispersion of dielectric thin films. *Appl. Phys. Lett.* **90**, 181112 (2007).
7. Harte, E., Alves I. D., Ihrke I. & Elezgaray J. Thickness determination in anisotropic media with plasmon waveguide resonance imaging. *Opt. Express* **27**, 3264-3275 (2019).
8. Beauzamy, L., Derr J. & Boudaoud A. Quantifying hydrostatic pressure in plant cells by using indentation with an atomic force microscope. *Biophys. J.* **108**, 2448-2456 (2015).
9. Wang, S., Boussaad S. & Tao N. J. Surface plasmon resonance enhanced optical absorption spectroscopy for studying molecular adsorbates. *Rev. Sci. Instrum.* **72**, 3055-3060 (2001).
10. Wang, T. et al. Scattering of electrically excited surface plasmon polaritons by gold nanoparticles studied by optical interferometry with a scanning tunneling microscope. *Phys. Rev. B* **92**, 045438 (2015).
11. Rothenhausler, B. & Knoll W. Surface-plasmon microscopy. *Nature* **332**, 615-617 (1988).
12. Banville, F. A., Moreau J., Sarkar M., Besbes M., Canva M. & Charette P. G. Spatial resolution versus contrast trade-off enhancement in high-resolution surface plasmon resonance imaging (SPRI) by metal surface nanostructure design. *Opt. Express* **26**, 10616-10630 (2018).
13. Watanabe, K., Matsuura K., Kawata F., Nagata K., Ning J. & Kano H. Scanning and non-scanning surface plasmon microscopy to observe cell adhesion sites. *Biomed. Opt. Express* **3**, 354-359 (2012).

14. Zhang, P., Ma G., Dong W., Wan Z., Wang S. & Tao N. Plasmonic scattering imaging of single proteins and binding kinetics. *Nat. Methods* **17**, 1010-1017 (2020).
15. Miyan, R. et al. Phase interrogation surface plasmon resonance hyperspectral imaging sensor for multi-channel high-throughput detection. *Opt. Express* **29**, 31418-31425 (2021).
16. Hruska, R., Mitchell J., Anderson M. & Glenn N. F. Radiometric and Geometric Analysis of Hyperspectral Imagery Acquired from an Unmanned Aerial Vehicle. *Remote Sensing* **4**, 2736-2752 (2012).
17. Jaud, M. et al. Easily Implemented Methods of Radiometric Corrections for Hyperspectral-UAV—Application to Guianese Equatorial Mudbanks Colonized by Pioneer Mangroves. *Remote Sensing* **13**, 4792 (2021).
18. Gupta, R. & Goddard N. J. Leaky waveguides (LWs) for chemical and biological sensing-A review and future perspective. *Sens. Actuators B Chem.* **322**, 128628 (2020).

REVIEWER COMMENTS

Reviewer #1 (Remarks to the Author):

The authors have addressed all the comments raised by this reviewer and the other reviewer. This manuscript brings SPR imaging innovation and expand SPR imaging applications. The experimental results in the revised manuscript validate their claims adequately. This manuscript satisfy the expectation of high impact of Nature Communications, and suitable for publication.

Reviewer #2 (Remarks to the Author):

The revision has addressed all of my questions in previous round of review, and I have no further questions.

September 23, 2022

Responses to Reviewers' comments:

Reviewer #1 (Remarks to the Author):

The authors have addressed all the comments raised by this reviewer and the other reviewer. This manuscript brings SPR imaging innovation and expand SPR imaging applications. The experimental results in the revised manuscript validate their claims adequately. This manuscript satisfy the expectation of high impact of Nature Communications, and suitable for publication.

Response: we greatly appreciate your favorable comments. Your first round of comments and inquiries have played an important role in our revision process of the manuscript. Thank you again for all your comments!

Reviewer #2 (Remarks to the Author):

The revision has addressed all of my questions in previous round of review, and I have no further questions.

Response: we greatly appreciate your constructive comments. Your first round of comments and inquiries have played an important role in our revision process of the manuscript. Thank you again for all your comments!